# The Power of Iterative Filtering for Supervised Learning with (Heavy) Contamination

**Adam R. Klivans**[*]
UT Austin

**Konstantinos Stavropoulos**[†]
UT Austin

**Kevin Tian**[‡]
UT Austin

**Arsen Vasilyan**[§]
UT Austin

## Abstract

Inspired by recent work on learning with distribution shift, we give a general outlier removal algorithm called *iterative polynomial filtering* and show a number of striking applications for supervised learning with contamination: (1) We show that any function class that can be approximated by low-degree polynomials with respect to a hypercontractive distribution can be efficiently learned under bounded contamination (also known as *nasty noise*). This is a surprising resolution to a longstanding gap between the complexity of agnostic learning and learning with contamination, as it was widely believed that low-degree approximators only implied tolerance to label noise. (2) For any function class that admits the (stronger) notion of sandwiching approximators, we obtain near-optimal learning guarantees even with respect to heavy additive contamination, where far more than $1/2$ of the training set may be added adversarially. Prior related work held only for regression and in a list-decodable setting. (3) We obtain the first efficient algorithms for tolerant testable learning of functions of halfspaces with respect to any fixed log-concave distribution. Even the non-tolerant case for a single halfspace in this setting had remained open. These results significantly advance our understanding of efficient supervised learning under contamination, a setting that has been much less studied than its unsupervised counterpart.

## 1 Introduction

Dataset curation is a fundamental part of the training pipeline of modern machine learning models and often appears to be the bottleneck in obtaining models with improved performance [SKL17, LRB+21, BGMMS21, GIF+23]. One way to theoretically model this problem is to assume that the learner has access to a—potentially heavily—contaminated dataset and the goal is to learn a model that performs well on some clean underlying target distribution. While there has been tremendous recent progress for unsupervised learning with contamination [HM13, CSV17, KS17a, KS17b, BDLS17, DKS18b, HL18, CDG19, RY20, BDH+20, CMY20, BK21, IK22, ZS22, DKK+22a, DHPT24], relatively little is known for *supervised* learning with contamination, especially for binary classification.

Many efficient algorithms with strong error guarantees have been developed for *agnostic* learning, a special case of contamination where only the labels are adversarially corrupted [KKMS08, KOS08, BOW10, ABL17, DKS18a, DKTZ20b, DKK+22b]. Most of these guarantees, however, had seemed difficult to extend to the more challenging setting of contamination, where both labels and covariates can be adversarially corrupted. In this paper—building on recent work on robust learning and learning with distribution shift [DKS18a, GSSV24, KSV24c]—we give a general iterative polynomial filtering procedure that greatly expands the set of known positive results for learning binary classifiers from

---

[*]klivans@cs.utexas.edu.

[†]kstavrop@utexas.edu.

[‡]kjtian@cs.utexas.edu.

[§]arsenvasilyan@gmail.com.

contaminated datasets (see Tables 1, 2, and 4). In particular, we obtain the surprising conclusion that all known near-optimal error guarantees for agnostic learning (that can be achieved efficiently) can indeed be extended to the setting of contamination.

**Learning with Bounded Contamination.** The earliest works that explored learnability beyond label noise date back more than 30 years ago [Val85, KL93]. Since then, the problem has been studied in the context of learning with *malicious* [Val85, KL93, KLS09, She25] and *nasty* [BEK02, ABL17, DKS18a, GSSV24, KSV24c] noise. Here we focus on the harshest among these noise models, nasty noise, which we call bounded contamination (defined below), in line with recent work in robust learning (see [DK23, KSV24c]). In this model, the adversary is allowed to replace an arbitrarily chosen but bounded fraction of a clean dataset with arbitrary labeled datapoints.

**Definition 1.1** (Bounded Contamination (BC)). *Let $\mathcal{D}$ be some distribution over $\mathcal{X}$, $\eta \in (0,1)$ and $f : \mathcal{X} \to \{\pm 1\}$. We say that a set of samples $\bar{S}_{\mathrm{inp}}$ is generated by $(\mathcal{D}, f)$ with bounded contamination of rate $\eta$ if it is generated as follows for some $M \geq 1$.[5]*

   *1. First, a set $\bar{S}_{\mathrm{cln}}$ of $M$ i.i.d. examples of the form $(\mathbf{x}, f(\mathbf{x}))$, where $\mathbf{x} \sim \mathcal{D}$, is drawn.*

   *2. Then, an adversary receives $\bar{S}_{\mathrm{cln}}$, chooses at most $\eta M$ labeled examples in $\bar{S}_{\mathrm{cln}}$ and substitutes them with an equal number of arbitrary labeled examples $\bar{S}_{\mathrm{adv}}$ to form $\bar{S}_{\mathrm{inp}}$.*

The learner receives a dataset with bounded contamination, and the goal is to output a classifier that enjoys information-theoretically optimal error guarantees on the clean underlying target distribution. Achieving an error better than twice the contamination rate is, in general, impossible [BEK02].

**Definition 1.2** (BC-Learning). *An algorithm $\mathcal{A}$ is a BC-learner for $\mathcal{C} \subseteq \{\mathcal{X} \to \{\pm 1\}\}$ if on input $(\epsilon, \delta, \bar{S}_{\mathrm{inp}})$, where $\epsilon, \delta \in (0,1)$, and $\bar{S}_{\mathrm{inp}}$ is generated by $(\mathcal{D}, f)$ with bounded contamination $\eta$ for some distribution $\mathcal{D}$ over $\mathcal{X}$, some $f \in \mathcal{C}$ and $\eta \in [0,1)$, the algorithm $\mathcal{A}$ outputs some hypothesis $h : \mathcal{X} \to \{\pm 1\}$ such that with probability at least $1 - \delta$ over the clean examples in $\bar{S}_{\mathrm{inp}}$, and the randomness of $\mathcal{A}$:*

$$\mathbb{P}_{\mathbf{x} \sim \mathcal{D}}[f(\mathbf{x}) \neq h(\mathbf{x})] \leq 2\eta + \epsilon$$

*The sample complexity of $\mathcal{A}$ is the minimum number of examples required to achieve the above guarantee. Moreover, a distribution-specific BC-learner with respect to some distribution $\mathcal{D}^*$ over $\mathcal{X}$ is a BC-learner that is guaranteed to work only when $\mathcal{D} = \mathcal{D}^*$.*

Most of the computationally efficient algorithms for learning with bounded contamination provide suboptimal error guarantees and apply only to special concept classes [KLS09, ABL17, DKS18a]. Nevertheless, a recent line of works inspired by advances in learning with distribution shift [GKKM20, KSV24b, GSSV24] has given efficient algorithms with near-optimal guarantees for concept classes that admit sandwiching polynomial approximators [GSSV24, KSV24c]. In contrast, for agnostic learning (i.e., adversarial label noise), it is well known that the weaker notion of (non-sandwiching) approximating polynomials is sufficient [KKMS08], and there is strong evidence of its necessity [DSFT+14, DKPZ21].

Therefore, the following question naturally arises: *does the existence of low-degree approximating polynomials imply efficient learnability even with respect to bounded contamination?*

In Theorem 4.2, we give a positive answer to this question, thereby resolving a longstanding gap between the complexity of learning with bounded contamination and with adversarial label noise. This is particularly important as it implies exponential improvements for BC-learning of fundamental concept classes like intersections of halfspaces, monotone functions, and convex sets (see Table 2).

**Learning with Heavy Contamination.** Perhaps surprisingly, to our knowledge, binary classification beyond bounded contamination is completely unexplored. In contrast, there is a substantial body of work for learning from datasets where only a small proportion comes from the clean distribution in unsupervised settings [CSV17, DKS18b, RY20, BK21, IK22, ZS22, DHPT24] and linear regression [RY20, KKK19, DHPT24]. These works typically provide list-decodable guarantees (i.e., multiple candidate hypotheses) or require access to a small trusted clean sample. Here, we define a new model for learning binary classifiers with heavy additive contamination that outputs a single hypothesis with a strong error guarantee under the clean distribution. Our model is inspired by recent work on regression (a basic supervised learning task) in additive semi-random models [JLM+23, KLL+23].

---

[5] We use the notation $\bar{S}$ to denote a labeled dataset and distinguish it from its unlabeled counterpart $S$.

**Definition 1.3** (Heavily Contaminated (HC) Datasets). *Let $\bar{\mathcal{D}}$ be some distribution over $\mathcal{X} \times \{\pm 1\}$ (we think of $\bar{\mathcal{D}}$ as the clean or uncorrupted distribution). We say that a set of samples $\bar{S}_{\text{inp}}$ is generated by $\bar{\mathcal{D}}$ with Q-heavy contamination if it is generated as follows for some $m \leq M$ with $M/m \leq Q$.*

1. *First, a set $\bar{S}_{\text{cln}}$ of $m$ i.i.d. labeled examples from $\bar{\mathcal{D}}$ is drawn.*

2. *Then, an adversary receives $\bar{S}_{\text{cln}}$ and adds $M - m$ arbitrary labeled examples to form $\bar{S}_{\text{inp}}$.*

The heavy contamination model only allows the adversary to add points, since removing an arbitrary fraction of the clean samples would correspond to learning with truncation [DGTZ18, DGTZ19, KZZ24], which is beyond the scope of this work (see also Remark G.2). Another difference between Definition 1.1 and Definition 1.3 is that in the HC model, clean labels need not be realized by some function in the given concept class. We instead consider the following quantity

$$\mathsf{opt}_{\text{total}} = \min_{f \in \mathcal{C}} \frac{1}{|\bar{S}_{\text{inp}}|} \sum_{(\mathbf{x},y) \in \bar{S}_{\text{inp}}} \mathbb{1}\{y \neq f(\mathbf{x})\} \,, \tag{1.1}$$

which is the minimum error achievable by the concept class $\mathcal{C}$ on the whole (contaminated) input dataset $\bar{S}_{\text{inp}}$ (including the misclassification errors on the clean samples). The error benchmark we consider is a rescaling of $\mathsf{opt}_{\text{total}}$, proportional to the heavy contamination ratio $Q$.

**Definition 1.4** (HC-Learning). *An algorithm $\mathcal{A}$ is an HC-learner for $\mathcal{C} \subseteq \{\mathcal{X} \to \{\pm 1\}\}$ if on input $(\epsilon, \delta, Q, \bar{S}_{\text{inp}})$, where $\epsilon, \delta \in (0,1)$, $Q \geq 1$ and $\bar{S}_{\text{inp}}$ is a Q-heavily contaminated set of labeled examples generated by distribution $\bar{\mathcal{D}}$ (as described in Definition 1.3), the algorithm $\mathcal{A}$ outputs some hypothesis $h : \mathcal{X} \to \{\pm 1\}$ such that with probability at least $1 - \delta$ over the clean examples in $\bar{S}_{\text{inp}}$, and the randomness of $\mathcal{A}$:*

$$\mathbb{P}_{(\mathbf{x},y) \sim \bar{\mathcal{D}}}[y \neq h(\mathbf{x})] \leq Q \cdot \mathsf{opt}_{\text{total}} + \epsilon \,, \text{ where } \mathsf{opt}_{\text{total}} \text{ is given by Eq. (1.1)}$$

*The (clean) sample complexity of $\mathcal{A}$ is the minimum number of clean examples $\bar{S}_{\text{inp}}$ needs to contain in order to achieve the above guarantee. Moreover, a distribution-specific HC-learner with respect to $\mathcal{D}^*$ is an HC-learner that is guaranteed to work only when the marginal of $\bar{\mathcal{D}}$ on $\mathcal{X}$ is $\mathcal{D} = \mathcal{D}^*$.*

We show that the dependence on $\Omega(Q \cdot \mathsf{opt}_{\text{total}})$ is, in fact, necessary, even if the clean labels are realized by the learned class $\mathcal{C}$. The quantity $Q \cdot \mathsf{opt}_{\text{total}}$ equals the number of errors $|\bar{S}_{\text{inp}}| \cdot \mathsf{opt}_{\text{total}}$ of the optimal classifier on the input set $\bar{S}_{\text{inp}}$ divided by the size of the clean dataset. Our lower bound essentially shows the existence of a contamination strategy that forces any HC learner to pay for all the mistakes of the optimal classifier on $\bar{S}_{\text{inp}}$, even if these are not made on the clean dataset. This is possible as the learner does not know which subset of $\bar{S}_{\text{inp}}$ is clean. In the following, we let $\mathsf{opt}_{\text{clean}} := \min_{f \in \mathcal{C}} \mathbb{P}_{(\mathbf{x},y) \sim \bar{\mathcal{D}}}[y \neq f(\mathbf{x})]$ be the optimum error under the clean distribution.

**Proposition 1.5** (Informal, see Propositions B.2 and B.3). *Let $\mathcal{C}$ be any non-trivial class. Then, no HC-learner for $\mathcal{C}$ can guarantee error better than $\frac{1}{2} \cdot Q \cdot \mathsf{opt}_{\text{total}}$, even when the clean distribution is realizable, i.e., $\mathsf{opt}_{\text{clean}} = 0$. Moreover, if $|\mathcal{X}| < \infty$, then no HC-learner for $\mathcal{C}_{\text{all}} = \{\pm 1\}^{\mathcal{X}}$ can guarantee error better than $Q \cdot \mathsf{opt}_{\text{total}}$, even when $\mathsf{opt}_{\text{clean}} = 0$.*

Naturally, one might wonder whether the existence of approximating polynomials is sufficient for HC-learning. In Theorem E.2 we give a negative answer to this question, by providing a lower bound on the sample complexity of HC-learning of monotone functions, which admits low-degree approximators. A recent line of works has used the stronger notion of sandwiching approximators to provide efficient algorithms for various challenging learning tasks [GKK23, KSV24b, GSSV24, CKK+24b, KSV24c]. Here, we expand on this paradigm and show that the existence of low-degree sandwiching polynomials implies efficient HC-learning as well (Theorem 4.4).

## 1.1 Our Results

**Learning from Contaminated Datasets.** In Table 1, we present an overview of the upper bounds we obtain as applications of our main theorems on learning with contamination (Theorems 4.2 and 4.4). For comparison, we also provide the corresponding results on learning with label noise from prior work. For the case of bounded contamination, and for constant error and confidence parameters $(\epsilon, \delta)$, our results match the best known bounds for learning with label noise, since in both cases, the existence of low-degree polynomial approximators is sufficient. See Tables 2 and 4 for further details.

| Concept Class | Target Marginal | Adversarial Label Noise *(Prior work)* | Bounded Contamin. *(This work)* | 2-Heavy Contamin. *(This work)* |
|:---:|:---:|:---:|:---:|:---:|
| Intersections of $k$ Halfspaces | $\mathcal{N}(0, \mathbf{I}_d)$ | $d^{O(\log k)}$ | $d^{O(\log k)}$ | $d^{O(k^6)}$ |
| Depth-$t$, Size-$d$ Boolean Circuits | $\mathrm{Unif}\{\pm 1\}^d$ | $d^{O(\log d)^{t-1}}$ | $d^{O(\log d)^{t-1}}$ | $d^{O(\log d)^{O(t)}}$ |
| Degree-$k$ PTFs | $\mathcal{N}(0, \mathbf{I}_d)$ | $d^{O(k^2)}$ | $d^{O(k^2)}$ | $d^{O_k(1)}$ |
| Monotone Functions | $\mathrm{Unif}\{\pm 1\}^d$ | $2^{\tilde{O}(\sqrt{d})}$ | $2^{\tilde{O}(\sqrt{d})}$ | $2^{\Omega(d)}$ |

Table 1: Bounds on the time complexity of learning under different noise models up to excess error $\epsilon = 0.1$ and failure probability $\delta = 0.01$. Except for the lower bound for monotone functions under heavy contamination, all remaining entries correspond to upper bounds. See Appendix D for more details on the complexity of learning with adversarial label noise.

All of our results work in the distribution-specific setting. Distributional assumptions are unavoidable, as there is strong evidence that distribution-free learning of even the simplest classes (e.g., linear classifiers) in the presence of noise is computationally hard [FGKP06, Dan16]. Although we present our results for specific standard target marginal distributions, our results hold for any hypercontractive (Definition 3.1) marginal $\mathcal{D}^*$ that can be sampled efficiently, as long as the degree of approximation (resp. sandwiching for HC-learning) of the learned class is low under $\mathcal{D}^*$. Hypercontractive distributions are an extremely wide class of probability distributions, which includes Gaussians, all log-concave distributions over $\mathbb{R}^d$, as well as product distributions over $\{\pm 1\}^d$. See Appendix C for an overview of relevant results in approximation theory.

On the lower bound side, we show that learning monotone functions with heavy contamination requires exponentially many samples, thereby separating bounded and heavy contamination. For a more thorough discussion on lower bounds for learning with contamination, see Appendix E.[6]

**Testable Learning.** A recent line of works in learning theory has focused on providing learning algorithms that can verify their distributional assumptions [RV23, GKK23, GKSV23, DKK+23, GKSV24, STW24, GSSV24]. These algorithms are allowed to either accept and output a classifier with certified optimal performance, or detect a violation of their target distributional assumptions and reject. Here, we provide improved results for a tolerant version of this problem where the algorithm has to accept even if the input distribution is close to the target distribution. Tolerant testable learning was first studied in [GSSV24] (see Definition H.1).

In Theorem H.2, we show that the existence of low-degree sandwiching polynomials implies efficient tolerant testable learning. Previously, [GSSV24] required the existence of low-degree $\mathcal{L}_2$-sandwiching polynomials, which is a stronger assumption. Moreover, all of the previous results in testable learning—even the non-tolerant variants—required that the sandwiching polynomials have bounded coefficients. Here, we do not impose such a requirement and obtain the first testable learning results for functions of halfspaces with respect to any fixed log-concave distribution. All prior work either gave worse error guarantees [GKSV24, GKSV23], or required target marginals with strictly sub-exponential tails [GKK23]. See Appendix H for more details.

## 1.2 Our Techniques

**Iterative Polynomial Filtering.** All of our results use the same iterative polynomial filtering algorithm of Theorem 3.2 with appropriate hyperparameter choices. The algorithm receives a set $S_{\mathrm{inp}}$ of data points and filters it, outputting a subset $S_{\mathrm{filt}}$ of $S_{\mathrm{inp}}$ that satisfies the following two conditions. First, any low-degree polynomial $p$ whose absolute expectation $\mathbb{E}[|p(\mathbf{x})|]$ over the pre-specified target

---

[6]We note that although learning with bounded contamination is clearly more challenging than learning with adversarial label noise, many of the known lower bounds for agnostic learning do not transfer directly to BC-learning as formalized in Definition 1.1 for technical reasons. See Appendix E for a way to circumvent this.

distribution $\mathcal{D}^*$ is small, will also have bounded average over $S_{\text{filt}}$, i.e. $\frac{1}{|S_{\text{filt}}|} \sum_{\mathbf{x} \in S_{\text{filt}}} p(\mathbf{x})$ is small. Second, if $S_{\text{inp}}$ contains a set of points $S$ that were initially generated independently by $\mathcal{D}^*$, then only a limited number of points from $S$ can be removed by the filtering (the allowed number of removed points is controlled by an appropriate hyperparameter).

Our algorithm is a refined version of bounded-degree outlier removal procedures from robust learning and learning with distribution shift [DKS18a, DKK$^+$19, GSSV24, KSV24c]. The general principle behind these algorithms is that one can iteratively find polynomials that violate the desired condition over the input set and use them to filter the input points. In particular, the algorithm removes the points that give such polynomials values larger than a threshold. By choosing this threshold appropriately, one can control the proportion of removed points that lie in the clean set $S$ in each step.

Prior work only gave guarantees for squared polynomials [DKS18a, GSSV24] and for non-negative polynomials [KSV24c]. Here, we give a filtering algorithm that preserves the expectation of any polynomial, as long as its absolute expectation is small with respect to the target distribution. This is crucial for our application in BC-learning. Moreover, our filtering procedure works for any hypercontractive target distribution $\mathcal{D}^*$, which is also true for [DKS18a, GSSV24] but not for [KSV24c], which only works for the uniform distribution over the hypercube.

**Learning with Bounded Contamination.** Our algorithm follows a two-phase approach: (1) Run our outlier-removal algorithm, and obtain a filtered subset $S_{\text{filt}}$. (2) Following an approach similar to [KKMS08], construct a predictor based on the polynomial $\widehat{p}$ with smallest $\mathcal{L}_1$ error on $S_{\text{filt}}$. We now give a sketch of the analysis of this algorithm and explain how an error of $O(\eta + \epsilon)$ can be guaranteed. (The optimal dependence of $2\eta + \epsilon$ is obtained by carefully refining the analysis below.)

What would happen if we ran the phase (2) without filtering the dataset beforehand? As shown in [KKMS08], this approach works in the agnostic setting, i.e. when an adversary can corrupt only the labels but not the examples. A key observation in [KKMS08] is that if $p^*$ is an $\epsilon$-approximating polynomial for the ground truth $f^*$, then after $\eta$ fraction of data labels are corrupted, the polynomial $p^*$ will have an $\mathcal{L}_1$-error of only at most $\eta + \epsilon$. However, in the more challenging setting of learning with contamination over $\mathbb{R}^d$, even a single corrupted data-point can cause the $\mathcal{L}_1$-error of $p^*$ to be arbitrarily large. The reason is that any non-zero polynomial over $\mathbb{R}^d$ will be arbitrarily large in absolute value when evaluated at inputs $\mathbf{x}$ far enough from the origin.

Hence, the first phase of our algorithm aims to filter out such bad input datapoints. The following basic observation is key to our approach: if $p^*$ is an $\epsilon^4$-approximator in $\mathcal{L}_2$ norm for a $\{\pm 1\}$-valued function $f^*$, then the average $\mathbb{E}_{\mathbf{x} \sim \mathcal{D}^*}[|(p^*(\mathbf{x}))^2 - 1|]$ is at most $O(\epsilon^2)$. This observation, together with our filtering guarantee, ensures that the average $\mathbb{E}_{\mathbf{x} \sim S_{\text{filt}}}[(p^*(\mathbf{x}))^2 - 1]$ is likewise bounded by $O(\epsilon)$ while removing almost exclusively outliers.

Yet, the set $S_{\text{filt}}$ might still contain many outliers. We show that the condition $\mathbb{E}_{\mathbf{x} \sim S_{\text{filt}}}[(p^*(\mathbf{x}))^2 - 1] \leq O(\epsilon)$ implies that the remaining outliers are not dangerous when it comes to phase (2) of our algorithm. Indeed, this condition tells us that the number of remaining outliers $\mathbf{x}$ in $S_{\text{filt}}$ with $|p^*(\mathbf{x})| > \tau$ is at most $O(|S_{\text{filt}}|\epsilon/\tau^2)$. Taking $\tau = 2$, we see that the total contribution to the $\mathcal{L}_1$ error of $p^*$ on $S_{\text{filt}}$ of outliers $\mathbf{x}$ with $|p^*(\mathbf{x})| > 2$ is $O(\epsilon)$. The remaining outliers contribute at most $O(\eta)$ to this error, since there are at most $O(\eta)$ of such outliers and each satisfies $|p^*(\mathbf{x})| \leq 2$.

Overall, we see that the $\mathcal{L}_1$ error of $p^*$ on $S_{\text{filt}}$ is at most $O(\eta + \epsilon)$, and therefore the polynomial $\widehat{p}$ found in phase 2 of our algorithm will also have an $\mathcal{L}_1$ error of at most $O(\eta + \epsilon)$ on $S_{\text{filt}}$. Since phase (1) only removed at most $O(\epsilon)$ clean datapoints, we conclude that the $\mathcal{L}_1$ error of $\widehat{p}$ on the clean dataset $S_{\text{cln}}$ is likewise $O(\eta + \epsilon)$, which we use to bound the out-of-distribution error of $\widehat{p}$.

**Learning with Heavy Contamination.** Our heavy contamination algorithm follows the same structure as the one for bounded contamination: we first run the iterative filtering algorithm and then run $\mathcal{L}_1$ polynomial regression on the filtered dataset. This time, however, we choose the hyperparameters of the iterative filtering algorithm so that the proportion of removed points that are clean is inversely proportional to the heavy contamination ratio $Q$. In this way, we make sure to remove only a small fraction of the clean points. To conclude the proof, we use the notion of sandwiching polynomials. In particular, if $f^* \in \mathcal{C}$ is the optimum classifier on the input dataset, and $p_{\text{up}}, p_{\text{down}}$ are two low-degree polynomials such that (1) $p_{\text{up}}(\mathbf{x}) \geq f^*(\mathbf{x}) \geq p_{\text{down}}(\mathbf{x})$ for all $\mathbf{x}$ and (2) $\mathbb{E}_{\mathbf{x} \sim \mathcal{D}^*}[p_{\text{up}}(\mathbf{x}) - p_{\text{down}}(\mathbf{x})] \leq O(\epsilon^2/Q)$, then the filtering process guarantees that $\mathbb{E}_{\mathbf{x} \sim S_{\text{filt}}}[p_{\text{up}}(\mathbf{x}) - p_{\text{down}}(\mathbf{x})]$ scales proportionally to $\epsilon$. Overall, this implies that $p_{\text{down}}$ has low $\mathcal{L}_1$-error under $S_{\text{filt}}$ and that $\mathcal{L}_1$-polynomial regression achieves near-optimal error guarantees.

## 1.3 Related Work

**Supervised Learning with Noise.** The majority of existing works on robust supervised learning focus on label noise. In order to obtain efficient algorithms, it is standard and often necessary to make assumptions on the marginal distribution [Dan16, DK22, DKMR22], although there have been recent attempts to relax those assumptions [CKK+24a]. Even under common distributional assumptions, the best possible error guarantees often require exponential dependence on the excess error parameter $\epsilon$ [DSFT+14, DKPZ21]. A line of works was focused on providing faster learning algorithms for some classes at the expense of relaxed error guarantees [ABL17, DKS18a, DKTZ20b] or under restricted noise models [DKTZ20a, DKK+22b]. We consider the more challenging scenario of learning with contamination, where there is noise on both the input examples and their labels. Before this work, efficient algorithms for learning with bounded contamination up to optimal error were known only for classes with low sandwiching degree [GSSV24, KSV24c], and nothing was known about classification under heavy contamination.

**Semi-Random Models.** Our Definition 1.3 is inspired by semi-random models, which lie between the average and worst case settings [BS95, FK01]. In particular, it resembles an instantiation of the semi-random model framework known as a *monotone adversary*, which breaks a statistical assumption used by a learning algorithm (e.g., i.i.d. draws from a known distribution) by providing additional data. Our approach is inspired by recent algorithms for supervised regression problems, e.g., solving linear systems, sparse recovery, or matrix completion [CG18, KLL+23, JLM+23, KLL+24], that are tolerant to monotone adversaries. These algorithms also use reweightings that come with certificates of success. However, a major qualitative difference between the aforementioned works and ours is that our Definition 1.3 does not require that the adversary uses labels consistent with a "clean hypothesis." Instead, our algorithms can tolerate label noise (alongside covariate noise) and achieve the information-theoretically optimal clean error under such a contamination model (Proposition 1.5).

**Testable Learning.** In recent years, a number of works has focused on verifying the assumptions of learning algorithms. Testable learning was introduced by [RV23] in the context of verifying the distributional assumptions of agnostic learners and there are several subsequent works on this setting [GKK23, GKSV24, DKK+23, GKSV23, DKLZ24, STW24]. This paradigm has since been expanded to testing for distribution shifts that may harm the performance of supervised learning algorithms [KSV24b, KSV24a, CKK+24b, CKLS25], or even testing noise assumptions [GKSV25]. Here, we study a tolerant version of testable agnostic learning that was first studied by [GSSV24], and provide the first guarantees for halfspaces with respect to any fixed log-concave measure.

Learning with heavy contamination can be thought of as a finding version of the testable learning problem. More specifically, in testable learning the goal is to decide whether the input dataset is structured enough so that a near-optimal hypothesis can be found efficiently, while in HC-learning the goal is to *find* a subset of the input that is structured enough. An analogous connection was observed in [GSSV24] between TDS learning [KSV24b] and PQ learning [GKKM20] in the context of learning under distribution shift. There, the goal was to either decide whether the (unlabeled) test examples come from a distribution that is similar to the one the learner has trained on (TDS), or find a subset of the test examples where the learner is confident in its predictions (PQ). See Remark G.1.

## 2 Notation

We consider a $d$-dimensional feature space $\mathcal{X}$ which will either be $\mathbb{R}^d$ or the hypercube $\{\pm 1\}^d$. A polynomial $p$ over $\mathbb{R}^d$ is of the form $p(\mathbf{x}) = \sum_{\alpha \subseteq \mathbb{N}^d} c_p(\alpha) \mathbf{x}^\alpha$, where $\mathbf{x}^\alpha = \prod_{i \in [d]} x_i^{\alpha_i}$ is a monomial of degree $\|\alpha\|_1$. The degree of $p$ is equal to the maximum $\|\alpha\|_1$ such that $c_p(\alpha) \neq 0$. Over $\{\pm 1\}^d$ we use the multilinear expansion $p(\mathbf{x}) = \sum_{\mathcal{I} \subseteq [d]} c_p(\mathcal{I}) \mathbf{x}^\mathcal{I}$, where $\mathbf{x}^\mathcal{I} = \prod_{i \in \mathcal{I}} x_i$. In both cases, we call $\mathbf{c}_p$ the vector of coefficients. We consider sets of points $S$ to contain examples that are separate instances of the corresponding elements of $\mathcal{X}$. We denote with $\bar{S}$ the corresponding labeled set of examples. We use $\mathbf{x} \sim S$ to say that $\mathbf{x}$ is drawn uniformly from $S$. For a labeled distribution $\bar{\mathcal{D}}$ over $\mathcal{X} \times \{\pm 1\}$, $\mathcal{D}$ is the marginal on $\mathcal{X}$. In the following the distribution-specific algorithms have sample access to the unlabeled target distribution $\mathcal{D}^*$. We use $\mathcal{N}_d = \mathcal{N}(0, \mathbf{I}_d)$ to denote the standard $d$-dimensional Gaussian and $\mathrm{Unif}_d = \mathrm{Unif}\{\pm 1\}^d$ for the uniform over the $d$-dimensional hypercube.

# 3 Iterative Polynomial Filtering

We first define hypercontractivity, which is a crucial assumption for our filtering procedure.

**Definition 3.1** (Hypercontractivity). We say that a distribution $\mathcal{D}$ over $\mathcal{X}$ is $A$-hypercontractive with respect to polynomials for some $A \geq 1$ if for any polynomial $p$ over $\mathcal{X}$ and any $t \geq 2$ we have

1. $\mathbb{E}_{\mathbf{x} \sim \mathcal{D}}[|p(\mathbf{x})|^t] \leq (At)^{\ell t} \big( \mathbb{E}_{\mathbf{x} \sim \mathcal{D}}[|p(\mathbf{x})|] \big)^t$, where $\ell = \deg(p)$

2. The absolute value of any degree-1 monomial has finite expectation under $\mathcal{D}$.

Our main algorithmic tool is the following theorem, which gives a way to filter an arbitrary set of examples in order to preserve the expectations of polynomials whose expected absolute value under some hypercontractive target distribution is small. For the proof, see Appendix F.

**Theorem 3.2** (Iterative Filtering). *Let $\mathcal{D}^*$ be an $A$-hypercontractive distribution over a $d$-dimensional space $\mathcal{X}$. Consider parameters $\epsilon, \delta \in (0,1)$ and $R, \ell, m \geq 1$, and let $S_{\mathrm{inp}}$ be an arbitrary set of examples in $\mathcal{X}$ and $S_{\mathrm{ref}}$ a set of $m_{\mathrm{ref}}$ i.i.d. examples from $\mathcal{D}^*$. For a sufficiently large universal constant $C \geq 1$, if $m_{\mathrm{ref}} \geq R^2 \frac{(CAd)^{2\ell}}{\epsilon^3} (\log \frac{1}{\delta})^{4\ell+1}$, then Algorithm 1 on input $(S_{\mathrm{inp}}, S_{\mathrm{ref}}, m, \ell, R, \epsilon)$ runs in time $\mathrm{poly}(|S_{\mathrm{inp}}|, m_{\mathrm{ref}}, (d+1)^\ell)$ and outputs $S_{\mathrm{filt}} \subseteq S_{\mathrm{inp}}$ such that the following hold:*

1. *Let $S_{\mathrm{cln}}$ be any set of $m$ i.i.d. examples from $\mathcal{D}^*$ where $m \geq CR^2 \frac{(2A(d+1))^{2\ell}}{\epsilon^3} \log \frac{1}{\delta}$. Suppose that $S_{\mathrm{inp}}$ is formed by first removing an arbitrary fraction of points in $S_{\mathrm{cln}}$ and then adding any number of arbitrary points. Then, the algorithm removes a relatively small number of the examples in $S_{\mathrm{cln}}$ that appear in $S_{\mathrm{inp}}$:*

$$|(S_{\mathrm{cln}} \cap S_{\mathrm{inp}}) \setminus S_{\mathrm{filt}}| \leq \frac{1}{R} \cdot |S_{\mathrm{inp}} \setminus S_{\mathrm{filt}}| + \frac{\epsilon m}{2} \,, \text{ with probability at least } 1 - \delta \text{ over } S_{\mathrm{cln}}, S_{\mathrm{ref}}$$

2. *For any polynomial $p$ of degree at most $\ell$, and $\mathbb{E}_{\mathbf{x} \sim \mathcal{D}^*}[|p(\mathbf{x})|] \leq \frac{\epsilon}{4R}$ we have:*

$$\sum_{\mathbf{x} \in S_{\mathrm{filt}}} p(\mathbf{x}) \leq \epsilon m \,, \text{ with probability at least } 1 - \delta \text{ over } S_{\mathrm{ref}}$$

---

**Algorithm 1:** Iterative Polynomial Filtering

**Input:** $S_{\mathrm{inp}}$ set of $M$ points, $S_{\mathrm{ref}}$ set of $m_{\mathrm{ref}}$ points, $m, \ell \in \mathbb{N}$, $R \geq 1$, $\epsilon \in (0,1)$
**Output:** Set $S_{\mathrm{filt}} \subseteq S_{\mathrm{inp}}$.

**1** Let $\beta \leftarrow 2(2A)^{2\ell}$; $\gamma \leftarrow \frac{\epsilon}{2R}$; $B \leftarrow 4(d+1)^{\frac{\ell}{2}}(\frac{\beta}{\epsilon})^{\frac{1}{2}}$; $\Delta \leftarrow \frac{\epsilon}{2B}$;
**2** Let $\mathcal{P}$ denote the family of polynomials $p$ of degree at most $\ell$ for which we have:
   $\mathbb{E}_{\mathbf{x} \sim S_{\mathrm{ref}}}[|p(\mathbf{x})|] \leq \gamma$ and $\mathbb{E}_{\mathbf{x} \sim S_{\mathrm{ref}}}[(p(\mathbf{x}))^2] \leq \beta$;
**3** $S \leftarrow \{\mathbf{x} \in S_{\mathrm{inp}} : |p(\mathbf{x})| \leq B \text{ for all } p \in \mathcal{P}\}$;
**4** **for** $i = 0, 1, 2, \ldots, M$ **do**
**5**     Compute $p^*$ and $\lambda^*$ as follows.

$$p^* = \arg\max_{p \in \mathcal{P}} \sum_{\mathbf{x} \in S} p(\mathbf{x}) \quad \text{and} \quad \lambda^* = \frac{1}{m} \sum_{\mathbf{x} \in S} p^*(\mathbf{x})$$

**6**     **if** $\lambda^* \leq \epsilon$ **then** return $S_{\mathrm{filt}} \leftarrow S$;
**7**     **else**
**8**         Let $\tau^* \geq 0$ be the smallest value such that
           $\frac{|S|}{m} \mathbb{P}_{\mathbf{x} \sim S}[|p^*(\mathbf{x})| > \tau^*] \geq R \cdot \mathbb{P}_{\mathbf{x} \sim S_{\mathrm{ref}}}[|p^*(\mathbf{x})| > \tau^*] + \Delta$;
**9**         $S \leftarrow S \setminus \{\mathbf{x} \in S : |p^*(\mathbf{x})| > \tau^*\}$;
**10**     **end**
**11** **end**

---

The algorithm iteratively removes the points that give large values to polynomials that do not satisfy the stopping criterion of line 6. The hyperparameter $R$ determines how selective the filtering is: larger

values of $R$ imply that less points will be removed in each iteration, and the proportion of removed points that are clean is smaller. The price one has to pay for larger choices of $R$ is that the guarantee of part 2 holds for polynomials with smaller absolute expectation under the target distribution.

Our algorithm requires access to a set of reference samples from the target distribution $\mathcal{D}^*$ and uses them to restrict its attention to polynomials with the desired properties under $\mathcal{D}^*$. The stopping criterion ensures that upon completion all of these polynomials will have bounded expectations under the empirical distribution over the filtered set. The bound depends on the hyperparameters $\epsilon$ (target error) and $m$ (effective size), but not on the degree $\ell$ of the polynomials considered.

# 4 Applications

## 4.1 Learning with Bounded Contamination

We give results for BC-learning of any concept class that can be approximated by low-degree polynomials in $\mathcal{L}_2$ distance.

**Definition 4.1** (Polynomial Approximators). For $\epsilon \in (0,1)$, we say that a class $\mathcal{C} \subseteq \{\mathcal{X} \to \{\pm 1\}\}$ has $\epsilon$-approximate degree $\ell = \ell(\epsilon)$ with respect to some distribution $\mathcal{D}^*$ over $\mathcal{X}$ if for any $f \in \mathcal{C}$ there is a polynomial $p$ of degree at most $\ell(\epsilon)$ such that $\mathbb{E}_{\mathbf{x} \sim \mathcal{D}^*}[(f(\mathbf{x}) - p(\mathbf{x}))^2] \leq \epsilon$.

Our main result additionally requires that the target marginal distribution is hypercontractive. This assumption is inherited by Theorem 3.2.

**Theorem 4.2** (Polynomial Approximation implies BC-Learning). *Let $\epsilon, \delta \in (0,1)$ and $A \geq 1$. Let $\mathcal{D}^*$ be some $A$-hypercontractive distribution over a $d$-dimensional space and let $\mathcal{C}$ be a concept class whose $\frac{\epsilon^4}{C}$-approximate degree w.r.t. $\mathcal{D}^*$ is $\ell$ for some large enough universal constant $C \geq 1$. Then, there is an algorithm that $(\epsilon, \delta)$-learns $\mathcal{C}$ under bounded contamination with respect to $\mathcal{D}^*$ in time $\mathrm{poly}(A^\ell, (\log \frac{1}{\delta})^\ell, (d+1)^\ell, \frac{1}{\epsilon})$, and has (clean) sample complexity at most $\frac{1}{\epsilon^6} O(Ad \log(1/\delta))^{4\ell+1}$.*

In Table 2, we summarize some of the new results we obtain as corollaries of Theorem 4.2, combined with appropriate known bounds on the approximate degree (see Appendix C.1). We also present the previous state-of-the-art results for comparison. For monotone functions and convex sets, no non-trivial results were known before this work. For halfspace intersections, we obtain exponential improvements over prior work, and for polynomial threshold functions, we obtain the first near-optimal error bounds. For small-depth circuits, we obtain an improved dependence on the depth.

| Concept Class | Target Marginal | Runtime | Error | Reference |
|---|---|---|---|---|
| Intersections of $k$ Halfspaces | $\mathcal{N}_d$ or $\mathrm{Unif}_d$ | $d^{\tilde{O}(\log(k)/\epsilon^8)}$ | $2\eta + \epsilon$ | This work |
| | $\mathcal{N}_d$ or $\mathrm{Unif}_d$ | $d^{\tilde{O}(k^6/\epsilon^4)}$ | $4\eta + \epsilon$ | [GSSV24] |
| | $\mathcal{N}_d$ | $\left(\frac{dk}{\epsilon}\right)^{O(1)} + \left(\frac{k}{\epsilon}\right)^{O(k^2)}$ | $\tilde{O}(k^{\frac{4}{11}} \eta^{\frac{1}{11}}) + \epsilon$ | [DKS18a] |
| Degree-$k$ PTFs | $\mathcal{N}_d$ | $d^{\tilde{O}(k^2/\epsilon^8)}$ | $2\eta + \epsilon$ | This work |
| | $\mathcal{N}_d$ | $\mathrm{poly}(d^k, 1/\epsilon)$ | $\tilde{O}(k^2 \eta^{\frac{1}{1+k}}) + \epsilon$ | [DKS18a] |
| | $\mathrm{Unif}_d$ | $d^{(\log(1/\epsilon))^{\tilde{O}(k^2)}/\epsilon^8}$ | $2\eta + \epsilon$ | This work |
| Monotone Functions | $\mathrm{Unif}_d$ | $d^{\tilde{O}(\sqrt{d}/\epsilon^8)}$ | $2\eta + \epsilon$ | This work |
| Convex Sets | $\mathcal{N}_d$ | $d^{\tilde{O}(\sqrt{d}/\epsilon^8)}$ | $2\eta + \epsilon$ | This work |
| Depth-$t$, Size-$s$ Circuits | $\mathrm{Unif}_d$ | $d^{O(\log s)^{t-1} \log 1/\epsilon}$ | $2\eta + \epsilon$ | This work |
| | $\mathrm{Unif}_d$ | $d^{O(\log s)^{O(t)} \log 1/\epsilon}$ | $2\eta + \epsilon$ | [KSV24c] |

Table 2: Bounds on the time complexity of learning with bounded contamination of (unknown) rate $\eta \in (0,1)$ up to failure probability $\delta = 0.01$.

Our results nearly match the best known upper bounds for agnostic learning, albeit with a worse dependence on the excess error parameter $\epsilon$. In particular, to achieve excess error $\epsilon$, we require

$O(\epsilon^4)$-approximating polynomials, while for agnostic learning an $O(\epsilon^2)$-approximation suffices. Therefore, our results imply, for example, a runtime of $d^{\tilde{O}(\log(k)/\epsilon^8)}$ for BC-learning of $k$-halfspace intersections with respect to $\mathcal{N}_d$, but agnostic learning can be done in $d^{\tilde{O}(\log(k)/\epsilon^4)}$ [KOS08].

The proof of Theorem 4.2 is based on a delicate analysis of the error on the filtered dataset obtained by applying Theorem 3.2 and can be found in Appendix G.1. The filtering algorithm is used to preserve the following important property of any polynomial $p^*$ that approximates a boolean function:

$$\mathbb{E}[(p^*(\mathbf{x}))^2] \leq 1 + O(\epsilon) \tag{4.1}$$

We show that obtaining a filtered set that preserves this property is sufficient for learning with bounded contamination. To prove this, we crucially use the fact that the noise is bounded and that the right hand side of Eq. (4.1) is approximately equal to $1$ (rather than some larger constant). On the other hand, by applying part 2 of Theorem 3.2 on the polynomial $q(\mathbf{x}) = (p^*(\mathbf{x}))^2 - 1$, we are able to ensure the desired property for $p^*$ on the filtered set.

## 4.2 Learning with Heavy Contamination

For heavy contamination, our results are based on the stronger notion of sandwiching approximators from pseudorandomness [Baz09].

**Definition 4.3** (Sandwiching Approximators). For $\epsilon \in (0,1)$, we say that a class $\mathcal{C} \subseteq \{\mathcal{X} \to \{\pm 1\}\}$ has $\epsilon$-sandwiching degree $\ell = \ell(\epsilon)$ with respect to some distribution $\mathcal{D}^*$ over $\mathcal{X}$ if for any $f \in \mathcal{C}$ there are two polynomials $p_{\mathrm{up}}, p_{\mathrm{down}}$ of degree at most $\ell(\epsilon)$ such that:

1. $p_{\mathrm{down}}(\mathbf{x}) \leq f(\mathbf{x}) \leq p_{\mathrm{up}}(\mathbf{x})$ for all $\mathbf{x} \in \mathcal{X}$ and

2. $\mathbb{E}_{\mathbf{x} \sim \mathcal{D}^*}[p_{\mathrm{up}}(\mathbf{x}) - p_{\mathrm{down}}(\mathbf{x})] \leq \epsilon$

Once more, our main result requires hypercontractivity of the marginal distribution.

**Theorem 4.4** (Sandwiching implies HC-Learning). *Let $\epsilon, \delta \in (0,1)$ and $A, Q \geq 1$. Let $\mathcal{D}^*$ be some $A$-hypercontractive distribution over a $d$-dimensional space and let $\mathcal{C}$ be a concept class whose $\frac{\epsilon^2}{CQ}$-sandwiching degree w.r.t. $\mathcal{D}^*$ is $\ell$ for some large enough constant $C \geq 1$. Then, there is an algorithm that $(\epsilon, \delta, Q)$-HC learns $\mathcal{C}$ with respect to $\mathcal{D}^*$ in time $\mathrm{poly}(A^\ell, (\log(1/\delta))^\ell, (d+1)^\ell, Q/\epsilon)$, and has (clean) sample complexity at most $\frac{Q^2}{\epsilon^5} \cdot O(Ad)^{2\ell} \cdot \log \frac{1}{\delta}$.*

The proof of Theorem 4.4 uses once more Algorithm 1 to filter the input set, but this time we set the hyperparameter $R$ to a value that scales with the heavy contamination ratio $Q$. This ensures that we keep most of the clean points, because the filtering process is highly selective. Due to the choice of $R$, the required sandwiching degree scales with $Q$ as well. As a consequence, for many classes, our bounds are exponential in $Q$. After filtering, the approximation property of the sandwiching polynomials is preserved under the empirical distribution over the filtered set, and this is sufficient for efficient learnability of the optimum hypothesis on the filtered set, due to Theorem A.3 by [KKMS08]. For a complete proof of Theorem 4.4 and quantitative bounds for HC-learning of various classes, see Appendix G.2 and Table 4.

**Limitations and Future Work.** We lay the groundwork for a principled study of efficient learning algorithms in the presence of contamination. There are several interesting directions for future work: (1) *Universality:* All of our algorithms require sample access to the target unlabeled distribution $\mathcal{D}^*$. Relaxing this requirement and obtaining algorithms that work universally with respect to broad classes of distributions is an important question. This is known to be possible for several problems in learning with label noise [BOW10, ABL17, DKTZ20b], robust unsupervised learning [KS17a, KS17b, DHPT24], as well as testable learning [GKSV23, CKK+24b]. (2) *Improved Error Guarantees:* Our algorithms obtain error guarantees that are proven to be optimal in general. However, it is an open question whether one can achieve error $\eta + \epsilon$ for BC-learning or error $\frac{1}{2}(Q \cdot \mathrm{opt}_{\mathrm{total}} + \mathrm{opt}_{\mathrm{clean}})$ for HC-learning for special classes, perhaps by allowing the learner to output randomized hypotheses (also known as probabilistic concepts). (3) *Characterization of Efficient Learnability under Contamination:* We show that $\mathcal{L}_2$ polynomial approximation suffices for efficient BC-learning and $\mathcal{L}_1$ sandwiching suffices for efficient HC-learning. However, it is not clear whether efficient learnability in these models is completely characterized by these notions of approximation. For agnostic learning, $\mathcal{L}_1$ polynomial approximation is known to characterize efficient learnability for algorithms in the statistical query framework [DSFT+14, DKPZ21]. See also Appendix E.

## Acknowledgments and Disclosure of Funding

Adam Klivans was supported by NSF award AF-1909204 and the NSF AI Institute for Foundations of Machine Learning (IFML). Konstantinos Stavropoulos was supported by the NSF AI Institute for Foundations of Machine Learning (IFML), by scholarships from Bodossaki foundation and Leventis foundation, and by the Apple Scholars in AI/ML PhD fellowship. Arsen Vasilyan was supported by the NSF AI Institute for Foundations of Machine Learning (IFML).

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

## A Additional Tools

In our proofs, we make use of the following results for hypercontractive distributions.

**Lemma A.1** (Loewner Concentration, Lemma 3.8 in [CKLS25], Lemma B.1 in [GSSV24]). *Let $\mathcal{D}$ be some $A$-hypercontractive distribution in $d$ dimensions and let $S$ be a set of $m$ i.i.d. examples from $\mathcal{D}$ where $m \geq (CAd)^{2\ell}(\log \frac{1}{\delta})^{4\ell+1}$ for some sufficiently large universal constant $C \geq 1$, $\delta \in (0,1)$ and $A, \ell, d \geq 1$. Then, with probability at least $1 - \delta$ over $S$ the following condition holds.*

$$\frac{1}{2}\mathbb{E}_{\mathbf{x}\sim\mathcal{D}}[(p(\mathbf{x}))^2] \leq \mathbb{E}_{\mathbf{x}\sim S}[(p(\mathbf{x}))^2] \leq 2\mathbb{E}_{\mathbf{x}\sim\mathcal{D}}[(p(\mathbf{x}))^2]\,, \text{ for any } p \text{ of degree at most } \ell$$

**Lemma A.2.** *Let $\mathcal{D}$ be some $A$-hypercontractive distribution in $d$ dimensions for some $A \geq 1$. Let $S$ be a set of $m$ i.i.d. samples from $\mathcal{D}$ and $\mathcal{P}_\beta(S)$ be the family of polynomials of degree at most $\ell$ such that $\mathbb{E}_{\mathbf{x}\sim S}[(p(\mathbf{x}))^2] \leq \beta$, where $\ell, \beta, d \geq 1$. Then, for any $\epsilon, \delta \in (0,1)$, if $m \geq (CAd)^{2\ell}(\log \frac{1}{\delta})^{4\ell+1}$, and $B \geq \sqrt{2\beta/\epsilon} \cdot (d+1)^{\ell/2}$ the following holds with probability at least $1 - \delta$ over $S$:*

$$\mathbb{P}_{\mathbf{x}\sim\mathcal{D}}[|p(\mathbf{x})| \leq B\,, \text{ for all } p \in \mathcal{P}_\beta(S)] \geq 1 - \epsilon$$

*Proof of Lemma A.2.* We first apply Lemma A.1 to show that, with probability at least $1 - \delta$ over $S$, for any $p \in \mathcal{P}_\beta(S)$ we have $\mathbb{E}_{\mathbf{x}\sim\mathcal{D}}[(p(\mathbf{x}))^2] \leq 2\mathbb{E}_{\mathbf{x}\sim S}[(p(\mathbf{x}))^2] \leq 2\beta$.

Consider now the moment matrix $\mathbf{M} = \mathbb{E}_{\mathbf{x}\sim\mathcal{D}}[\psi_\ell(\mathbf{x})\psi_\ell(\mathbf{x})^\top]$, where $\psi_\ell(\mathbf{x})$ is the vector whose coordinates correspond to monomials of $\mathbf{x}$ of degree at most $\ell$. Let $\mathbf{M} = \mathbf{U}\mathbf{D}\mathbf{U}^\top$ be the concise SVD of $\mathbf{M}$, i.e., $\mathbf{D}$ is diagonal with dimension equal to the rank of $\mathbf{M}$, $\mathbf{U}^\top\mathbf{U} = \mathbf{I}$ and $\mathbf{U}\mathbf{U}^\top$ is the orthogonal projection on the column (or row) space of $\mathbf{M}$. Let $\varphi(\mathbf{x}) = \mathbf{D}^{-1/2}\mathbf{U}^\top\psi_\ell(\mathbf{x})$ and observe that $\mathbb{E}_{\mathbf{x}\sim\mathcal{D}}[\varphi(\mathbf{x})\varphi(\mathbf{x})^\top] = \mathbf{I}$ and the dimension of $\varphi$ is equal to the rank of $\mathbf{M}$, which is at most $(d+1)^\ell$. Here, we use the fact that all the entries of $\mathbf{M}$ are finite, due to hypercontractivity.

We may express any polynomial $p$ in $\mathcal{P}_\beta(S)$ as a linear combination of the elements of $\varphi$, i.e., $p(\mathbf{x}) = \tilde{\mathbf{c}}_p^\top\varphi(\mathbf{x})$, where $\|\tilde{\mathbf{c}}_p\|_2^2 = \mathbb{E}_{\mathbf{x}\sim\mathcal{D}}[(p(\mathbf{x}))^2] \leq 2\beta$. Moreover, we have the following due to Markov's inequality and the fact that $\mathbb{E}[\|\varphi(\mathbf{x})\|_2^2] \leq (d+1)^\ell$.

$$\mathbb{P}_{\mathbf{x}\sim\mathcal{D}}\Big[\|\varphi(\mathbf{x})\|_2^2 > \frac{B^2}{2\beta}\Big] \leq \frac{2\beta(d+1)^\ell}{B^2} \leq \epsilon$$

In the event that $\|\varphi(\mathbf{x})\|_2^2 \leq \frac{B^2}{2\beta}$, we have $|p(\mathbf{x})| \leq \|\tilde{\mathbf{c}}_p\|_2\|\varphi(\mathbf{x})\|_2 \leq \sqrt{2\beta} \cdot \frac{B}{\sqrt{2\beta}} = B$. $\qquad\square$

We will also use the following theorem which is implicit in [KKMS08] and shows that polynomial approximation is sufficient to find a simple hypothesis with near-optimum error on a given dataset.

**Theorem A.3** ($\mathcal{L}_1$ polynomial regression [KKMS08]). *Let $\bar{S}$ be a set of labeled examples and $\mathcal{C}$ be a concept class such that for each $f \in \mathcal{C}$ there is some polynomial $p$ of degree at most $\ell$ such that $\mathbb{E}_{\mathbf{x}\sim S}[|f(\mathbf{x}) - p(\mathbf{x})|] \leq \epsilon$. Then, the degree-$\ell$ $\mathcal{L}_1$ polynomial regression algorithm of [KKMS08] outputs, in time $\mathrm{poly}(|\bar{S}|d^\ell/\epsilon)$, a degree-$\ell$ polynomial threshold function $h$ such that $\mathbb{P}_{(\mathbf{x},y)\sim\bar{S}}[y \neq h(\mathbf{x})] \leq \min_{f\in\mathcal{C}}\mathbb{P}_{(\mathbf{x},y)\sim\bar{S}}[y \neq f(\mathbf{x})] + \epsilon$.*

We also give the definition of $\mathcal{L}_1$ approximating polynomials, which are sufficient for agnostic learning, but it is not clear whether they suffice for BC-learning. We instead use the notion of $\mathcal{L}_2$ approximating polynomials (Definition 4.1).

**Definition A.4** ($\mathcal{L}_1$ Polynomial Approximators). *For $\epsilon \in (0,1)$, we say that a class $\mathcal{C} \subseteq \{\mathcal{X} \to \{\pm 1\}\}$ has $\epsilon$-approximate degree $\ell = \ell(\epsilon)$ with respect to some distribution $\mathcal{D}^*$ over $\mathcal{X}$ if for any $f \in \mathcal{C}$ there is a polynomial $p$ of degree at most $\ell(\epsilon)$ such that $\mathbb{E}_{\mathbf{x}\sim\mathcal{D}^*}[|f(\mathbf{x}) - p(\mathbf{x})|] \leq \epsilon$.*

The difference between $\mathcal{L}_1$ and $\mathcal{L}_2$ approximators is that the former enjoy the approximation property with respect to the $\mathcal{L}_1$ norm, while the latter with respect to the $\mathcal{L}_2$ norm. Achieving $\mathcal{L}_2$ approximation is, in general, a stronger assumption, because the $\mathcal{L}_1$ norm is upper bounded by the $\mathcal{L}_2$ norm, but not vice versa. Nevertheless, for most interesting classes that $\mathcal{L}_1$ approximators are known, we also have $\mathcal{L}_2$ approximators. This is because of analytic advantages of the $\mathcal{L}_2$ norm.

# B Error Lower Bounds for HC-Learning

In order to prove our lower bounds, we provide a definition for the adversary in the context of heavy contamination.

**Definition B.1** (Contamination Strategy). An algorithm $\mathcal{M}$ is called a (randomized) $Q$-HC strategy if, upon receiving a labeled set $\bar{S}_{\text{cln}}$, it outputs a set $\bar{S}_{\text{inp}}$ of size $|\bar{S}_{\text{inp}}| \leq Q|\bar{S}_{\text{cln}}|$ such that $\bar{S}_{\text{inp}} \subseteq \bar{S}_{\text{cln}}$. For $m \in \mathbb{N}$ and a labeled distribution $\bar{\mathcal{D}}$, we denote with $\mathcal{M}(\bar{\mathcal{D}}^m)$ the distribution of the output $\bar{S}_{\text{inp}}$ of $\mathcal{M}$ on a set of $m$ i.i.d. samples $\bar{S}_{\text{cln}}$ from $\bar{\mathcal{D}}$. Note that $\bar{S}_{\text{inp}}$ depends on the random choice of $\bar{S}_{\text{cln}}$, as well as the randomness of $\mathcal{M}$.

We first prove the following lower bound, which holds for any non-trivial concept class.

**Proposition B.2.** *Let $\mathcal{C}$ be any concept class such that there is some $f \in \mathcal{C}$ for which $-f \in \mathcal{C}$. For any $Q \in \{2, 3, \dots\}$, any unlabeled distribution $\mathcal{D}^*$, any $\rho \in [0, 1/2)$, $\epsilon, \delta > 0$, and any $m \geq \frac{\log(2/\delta)}{2\epsilon^2}$, there is a $Q$-HC strategy $\mathcal{M}$ and a distribution $\bar{\mathcal{D}}$ whose marginal is $\mathcal{D}^*$ such that, if we let $\bar{\mathcal{D}}'$ be the distribution of $(\mathbf{x}, -y)$, where $(\mathbf{x}, y) \sim \bar{\mathcal{D}}$, then the following hold.*

1. *$\mathcal{M}(\bar{\mathcal{D}}^m) = \mathcal{M}((\bar{\mathcal{D}}')^m)$.*

2. *$\min_{f \in \mathcal{C}} \mathbb{P}_{(\mathbf{x}, y) \sim \bar{\mathcal{D}}}[y \neq f(\mathbf{x})] = \min_{f \in \mathcal{C}} \mathbb{P}_{(\mathbf{x}, y) \sim \bar{\mathcal{D}}'}[y \neq f(\mathbf{x})] = \rho$.*

3. *With probability at least $1 - \delta$ over $\bar{S}_{\text{inp}} \sim \mathcal{M}(\bar{\mathcal{D}}^m)$, we have $Q \cdot \mathsf{opt}_{\text{total}} \leq 1 - \rho + \epsilon$, where $\mathsf{opt}_{\text{total}} = \min_{f \in \mathcal{C}} \mathbb{P}_{(\mathbf{x}, y) \sim \bar{S}_{\text{inp}}}[y \neq f(\mathbf{x})]$.*

*Hence, no HC-learner for $\mathcal{C}$ with respect to $\mathcal{D}^*$ can achieve error less than $\frac{1}{2}(Q \cdot \mathsf{opt}_{\text{total}} + \mathsf{opt}_{\text{clean}} - \epsilon)$ with probability more than $\frac{1}{2} - \delta$, even when $\mathsf{opt}_{\text{clean}} = \rho$.*

*Proof of Proposition B.2.* Let $\mathcal{D} = \mathcal{D}^*$. Consider the following two labeled distributions. First, $\bar{\mathcal{D}}_1$ is a distribution over $\mathcal{X} \times \{\pm 1\}$ whose marginal on $\mathcal{X}$ is $\mathcal{D}$ and the labels are generated as follows:

$$y_1 = \begin{cases} f(\mathbf{x}), & \text{with probability } 1 - \rho \\ -f(\mathbf{x}), & \text{with probability } \rho \end{cases}$$

Consider also a distribution $\bar{\mathcal{D}}_2$ whose marginal is also $\mathcal{D}$ but the labels are generated as follows:

$$y_2 = \begin{cases} -f(\mathbf{x}), & \text{with probability } 1 - \rho \\ f(\mathbf{x}), & \text{with probability } \rho \end{cases}$$

The adversary receives a set $\bar{S}_{\text{cln}}$ of $m$ i.i.d. samples from $\bar{\mathcal{D}} \in \{\bar{\mathcal{D}}_1, \bar{\mathcal{D}}_2\}$ and does the following.

1. First, the adversary computes the value $m_1 = \sum_{(\mathbf{x}, y) \in \bar{S}_{\text{inp}}} \mathbb{1}\{y = f(\mathbf{x})\}$, as well as the value $m_2 = \sum_{(\mathbf{x}, y) \in \bar{S}_{\text{inp}}} \mathbb{1}\{y = -f(\mathbf{x})\}$.

2. If $m_1 \geq m_2$, then the adversary adds $M - m$ examples drawn by $\mathcal{D}$ and labeled by $-f(\mathbf{x})$.

3. If $m_1 < m_2$, then the adversary adds $m_2 - m_1$ examples drawn by $\mathcal{D}$ and labeled by $f(\mathbf{x})$, as well as $M - 2m_2$ examples drawn by $\mathcal{D}$ and labeled by $-f(\mathbf{x})$.

**Part 1.** Let $\xi = m_i$, where $i \in \{1, 2\}$ such that $\bar{\mathcal{D}} = \bar{\mathcal{D}}_i$. We have that $\xi$ follows the binomial distribution with parameters $(m, 1 - \rho)$, and is independent from the value of $i$.

Given any realization $m_i$ of $\xi$ with $m_i > m/2$, we can equivalently form the input examples $\bar{S}_{\text{inp}}$ by drawing $M$ i.i.d. samples from $\mathcal{D}$ and labeling $\xi$ of them according to $f(\mathbf{x})$ and $M - \xi$ of them according to $-f(\mathbf{x})$. We have that $\mathcal{M}(\bar{\mathcal{D}}_1^m)|_{\xi > m/2} = \mathcal{M}(\bar{\mathcal{D}}_2^m)|_{\xi > m/2}$.

Given any realization $m_i$ of $\xi$ with $m_i \leq m/2$, we can equivalently form the input examples $\bar{S}_{\text{inp}}$ by drawing $M$ i.i.d. samples from $\mathcal{D}$ and labeling $m - \xi$ of them according to $f(\mathbf{x})$ and $M - m + \xi$ of them according to $-f(\mathbf{x})$. We, therefore, have $\mathcal{M}(\bar{\mathcal{D}}_1^m)|_{\xi \leq m/2} = \mathcal{M}(\bar{\mathcal{D}}_2^m)|_{\xi \leq m/2}$.

Overall, since $\xi$ does not depend on $i$, we have $\mathcal{M}(\bar{\mathcal{D}}_1^m) = \mathcal{M}(\bar{\mathcal{D}}_2^m)$.

**Part 2.** Follows immediately from the fact that $f, -f \in \mathcal{C}$ and the definition of $\bar{\mathcal{D}} = \bar{\mathcal{D}}_1, \bar{\mathcal{D}}' = \bar{\mathcal{D}}_2$.

**Part 3.** Regarding the error benchmark, we have $Q \cdot \mathsf{opt}_{\mathrm{total}} = \frac{\max\{\xi, m-\xi\}}{m} \le 1 - \rho + \epsilon$, whenever $|\xi - (1-\rho)m| \le \epsilon m$, which happens with probability at least $1 - \delta$, according to the following inequality.

$$\mathbb{P}[|\xi - (1-\rho)m| > \epsilon m] \le 2 \exp(-2\epsilon^2 m) \le \delta$$

The bound follows from an application of Hoeffding's inequality, since $\xi$ is binomial of mean $(1-\rho)m$. It follows that with probability at least $1 - \delta$, the algorithm outputs a hypothesis with error at least $\frac{1}{2}(Q \cdot \mathsf{opt}_{\mathrm{total}} + \mathsf{opt}_{\mathrm{clean}} - \epsilon)$.

**Implication to HC Learning.** We have $\mathcal{M}(\bar{\mathcal{D}}_1^m) = \mathcal{M}(\bar{\mathcal{D}}_2^m)$. Let $\mathcal{A}$ be any, potentially randomized algorithm, and let $h := \mathcal{A}(\mathcal{M}(\bar{\mathcal{D}}_1^m)) = \mathcal{A}(\mathcal{M}(\bar{\mathcal{D}}_2^m))$ be the random variable corresponding to its output, where $h : \mathcal{X} \to \{\pm 1\}$. Consider the event that $h$ is such that: $\mathbb{P}_{(\mathbf{x},y) \sim \bar{\mathcal{D}}_1}[h(\mathbf{x}) \ne y] < 1/2$. Then, we have:

$$\mathbb{P}_{(\mathbf{x},y) \sim \bar{\mathcal{D}}_2}[h(\mathbf{x}) \ne -y] = \mathbb{P}_{(\mathbf{x},y) \sim \bar{\mathcal{D}}_1}[h(\mathbf{x}) \ne -y] = 1 - \mathbb{P}_{(\mathbf{x},y) \sim \bar{\mathcal{D}}_1}[h(\mathbf{x}) \ne y] > 1/2$$

Note that the same is true even if $h$ is a randomized classifier. Overall, the error of $\mathcal{A}$ on either $\bar{\mathcal{D}}_1$ or $\bar{\mathcal{D}}_2$ is $1/2$ with probability at least $1/2$. Finally, due to part 3, with probability at least $1 - \delta$, we have $1/2 \ge \frac{1}{2}(Q \cdot \mathsf{opt}_{\mathrm{total}} + \mathsf{opt}_{\mathrm{clean}} - \epsilon)$, which implies the desired result. $\qquad\square$

For finite feature spaces and the particular choice of the class of all functions, we obtain a sharper information-theoretic lower bound on the error. Note that this lower bound captures the problem of HC-learning halfspaces over $\{\pm 1\}$ in one dimension.

**Proposition B.3.** *Suppose that $|\mathcal{X}| < \infty$ and $\mathcal{C}$ is any concept class from $\mathcal{X}$ to $\{\pm 1\}$. Then, on any value of $Q$ in $\{2, 3, \dots\}$, any (distribution-specific) HC-learner will output a hypothesis with error $1$ with probability at least $2^{-|\mathcal{X}|}$ on some instance where $Q \cdot \mathsf{opt}_{\mathrm{total}} = 1$. Moreover, if $\mathcal{C} = \{\pm 1\}^{\mathcal{X}}$, then the instance satisfies $\mathsf{opt}_{\mathrm{clean}} = 0$.*

*Proof of Proposition B.3.* The adversary receives the clean dataset $\bar{S}_{\mathrm{cln}}$ and creates a duplicate $\bar{S}'_{\mathrm{cln}}$, where all the labels are flipped but the feature vectors are unchanged. Then, the adversary draws $(Q-2) \cdot m$ i.i.d. examples $\bar{S}'$ from the marginal distribution $\mathcal{D}$ on $\mathcal{X}$ of the clean distribution $\bar{\mathcal{D}}$ and labels them according to $f_1$. The input dataset is $\bar{S}_{\mathrm{cln}} \cup \bar{S}'_{\mathrm{cln}} \cup \bar{S}'$.

Note that the input dataset is completely independent from the clean labels, since $\bar{S}_{\mathrm{cln}} \cup \bar{S}'_{\mathrm{cln}}$ can be constructed equivalently by choosing an arbitrary function $g : \mathcal{X} \to \{\pm 1\}$ to label the points in $\bar{S}_{\mathrm{cln}}$ and then label the points in $\bar{S}'_{\mathrm{cln}}$ according to $-g$.

Therefore, we may assume that the ground truth labels are generated by the function $f^*$, where $-f^*$ is the most likely output of the HC-learner on a dataset of the form $\bar{S}_{\mathrm{cln}} \cup \bar{S}'_{\mathrm{cln}} \cup \bar{S}'$. Since there are at most $2^{|\mathcal{X}|}$ possible functions, $\mathcal{A}$ must output $-f^*$ with probability at least $2^{-|\mathcal{X}|}$. When $\mathcal{A}$ outputs $-f^*$, the error is $1$.

Finally, note that $f_1$ achieves error on $\bar{S}_{\mathrm{inp}}$ equal to $1/Q$, so $\mathsf{opt}_{\mathrm{total}} = 1/Q$ and $Q \cdot \mathsf{opt}_{\mathrm{total}} = 1$.

When $\mathcal{C} = \{\pm 1\}^{\mathcal{X}}$, we have $f^* \in \mathcal{C}$ and, hence, $\mathsf{opt}_{\mathrm{clean}} := \min_{f \in \mathcal{C}} \mathbb{P}_{(\mathbf{x},y) \sim \bar{\mathcal{D}}}[y \ne f(\mathbf{x})] = 0$. $\quad\square$

# C  Approximation Theory Results

## C.1  Low-Degree Approximators

Our structural Theorem 4.2 can be combined with results from polynomial approximation theory in order to obtain end-to-end results on learning with nasty noise, including the results of Table 2.

**Intersections of Halfspaces.** For the class of $k$-halfspace intersections, [KOS08] showed that the $\epsilon$-approximate degree with respect to the standard Gaussian distribution $\mathcal{N}_d$ (see Definition 4.1) is $\ell(\epsilon) \le \log(k)/\epsilon^2$ and [Kan14] obtains the same result with respect to the uniform distribution $\mathrm{Unif}_d$ over the hypercube.

**Monotone Functions and Convex Sets.** For monotone functions, the $\epsilon$-approximate degree with respect to $\mathrm{Unif}_d$ was bounded by $\ell(\epsilon) \leq \sqrt{d}/\epsilon$ in [BT96]. For convex sets, [KOS08] showed that the $\epsilon$-approximate degree with respect to the Gaussian distribution can be bounded via the Gaussian Surface Area (GSA). Combined with the result of [Bal93] on the GSA of convex sets, the corresponding bound is $\ell(\epsilon) \leq \sqrt{d}/\epsilon^2$.

**Polynomial Threshold Functions.** For the class of degree-$k$ PTFs, the $\epsilon$-approximate degree with respect to the standard Gaussian was bounded by $\ell(\epsilon) \leq k^2/\epsilon^2$ in [Kan11a]. In [Kan13], it was shown that $\ell(\epsilon) \leq \frac{k^{O(k^2)}(\log 1/\epsilon)^{O(k \log k)}}{\epsilon^2}$ with respect to the uniform over $\{\pm 1\}^d$.

**Low-depth Circuits.** For the class of $\mathsf{AC}^0$ circuits of size $s$ and depth $t$, the $\epsilon$-approximate degree is known to be at most $\ell(\epsilon) = O(\log(s))^{t-1} \log(1/\epsilon)$, due to the seminal work of [LMN93] and subsequent improvements [Tal17].

## C.2 Sandwiching Approximators

For the more challenging problem of learning with heavy contamination, we require the existence of sandwiching approximators (see Theorem 4.4), and show that arbitrary (non-sandwiching) approximators do not suffice (see Theorem E.2).

**Decision Trees of Halfspaces.** For the class of depth-$t$, size-$s$ decision trees of halfspaces, [GOWZ10] showed that the $\epsilon$-sandwiching degree with respect to either the standard Gaussian or the uniform on the hypercube is $\ell(\epsilon) \leq \tilde{O}(\frac{t^4 s^2}{\epsilon^2})$.

**Low-Depth Circuits.** The $\epsilon$-sandwiching degree of $\mathsf{AC}^0$ circuits of depth $t$ and size $s$ is known to be at most $\ell(\epsilon) \leq O(\log(s))^{O(t)} \log(1/\epsilon)$, due to the celebrated results of [Bra08, Tal17, HS19].

**Polynomial Threshold Functions.** The $\epsilon$-sandwiching degree of degree-$k$ PTFs over the Gaussian was bounded by $\ell(\epsilon) \leq O_k(\epsilon^{-4k \cdot 7^k})$ in [STW24], based on the pseudorandom generator of [Kan11b]. Here, $O_k(\cdot)$ is hiding a multiplicative factor that scales as an arbitrary function of $k$. For $k = 2$, the sandwiching degree with respect to the uniform distribution over the hypercube is known to be at most $\ell(\epsilon) \leq O(1/\epsilon^9)$ due to [DKN10].

**Functions of Halfspaces over Log-Concave Measures.** For arbitrary functions of $k$ halfspaces, the $\epsilon$-sandwiching degree with respect to any log-concave distribution was shown to be at most $\ell(\epsilon) \leq \exp((\log(\log(k)/\epsilon))^{O(k)}/\epsilon^4)$ by [KM13, KKM13].

# D Complexity of Learning with Adversarial Label Noise

The computational complexity of agnostic learning (or equivalently learning with adversarial label noise) is much better understood than the complexity of learning with contamination. In this setting, the learner receives i.i.d. examples from some labeled distribution $\bar{\mathcal{D}}$ whose marginal on the feature space $\mathcal{X}$ is well-behaved (i.e., Gaussian or uniform) and is, otherwise, arbitrary. The goal is to output, with high probability, a hypothesis whose error is at most $\mathsf{opt} + \epsilon$, where $\mathsf{opt} = \min_{f \in \mathcal{C}} \mathbb{P}_{(\mathbf{x},y) \sim \bar{\mathcal{D}}}[y \neq f(\mathbf{x})]$, for some target concept class $\mathcal{C}$.

**Upper Bounds.** On the upper bound side, [KKMS08] showed that any class with bounded $\mathcal{L}_1$ approximation degree admits dimension-efficient agnostic learners via low-degree $\mathcal{L}_1$ polynomial regression. Several prior and subsequent works provided such bounds for many fundamental classes, based on Fourier or Hermite analysis [LMN93, BT96, DHK+10, Kan11a, Kan13, Kan14], as well as geometric properties of Gaussian spaces [KOS08]. Note that the usual way to bound the $\mathcal{L}_1$ approximation degree, is to first bound the $\mathcal{L}_2$ approximation degree and then use Cauchy-Schwarz inequality. This is because bounding the $\mathcal{L}_2$ approximation is usually more analytically convenient. However, this approach does not always give tight results for the degree in terms of the approximation error. In [FKV20], it was shown that $\mathcal{L}_1$ approximation bounds over the hypercube can be obtained directly for any function class with bounded noise sensitivity.

| Concept Class | Target Marginal | Upper Bounds | Lower Bounds |
|---|---|---|---|
| Intersections of $k$ Halfspaces | $\mathcal{N}(0, \mathbf{I}_d)$ | $d^{\tilde{O}(\log(k)/\epsilon^4)}$ | $d^{\Omega(\sqrt{\log k}/\epsilon)}$ |
| Depth-$t$, Size-$d$ Boolean Circuits | $\mathrm{Unif}\{\pm 1\}^d$ | $d^{O(\log d)^{t-1}\cdot \log 1/\epsilon}$ | $d^{(\log d)^{\Omega(t)}} (\epsilon = O(1))$ |
| Degree-$k$ PTFs | $\mathcal{N}(0, \mathbf{I}_d)$ | $d^{\tilde{O}(k^2/\epsilon^4)}$ | $d^{\Omega(k^2/\epsilon^2)}$ |
| Monotone Functions | $\mathrm{Unif}\{\pm 1\}^d$ | $2^{\tilde{O}(\sqrt{d}/\epsilon^2)}$ | $2^{\Omega(\sqrt{d}/\epsilon)}$ |

Table 3: Upper and lower bounds on the time complexity of agnostic learning up to excess error $\epsilon$ and failure probability $\delta = 0.01$. The lower bounds either hold for algorithms in the Statistical Query model [Kea98] or for any algorithm under appropriate cryptographic assumptions.

**Lower Bounds.** The complexity of agnostic learning with respect to Gaussian marginals was shown to be characterized in terms of the $\mathcal{L}_1$ approximation degree for algorithms that fall in the statistical query (SQ) framework by [DKPZ21], implying the lower bounds of Table 3 for intersections of halfspaces and polynomial threshold functions. For the uniform distribution on the hypercube, a similar characterization was given by [DSFT+14]. Moreover, [BCO+15] showed an information-theoretic lower bound for learning monotone functions even in the realizable setting, implying the lower bound of Table 3 for monotone functions. For AC$^0$ circuits, [Kha93] provided a cryptographic lower bound.

# E  Lower Bounds for Learning with Contamination

**Bounded Contamination.** Although the problem of learning from contaminated datasets is, in principle, more challenging than learning with label noise, it is not clear whether the common formulations of the former are formally stronger than those of the latter. In particular, the problem of learning with label noise is usually formalized in terms of agnostic learning, where the goal is to achieve error $\mathsf{opt} + \epsilon$ on the *input* distribution $\bar{\mathcal{D}}$ for $\mathsf{opt} = \min_{f \in \mathcal{C}} \mathbb{P}_{(\mathbf{x},y)\sim\bar{\mathcal{D}}}[y \neq f(\mathbf{x})]$, where $\mathcal{C}$ is the learned class. In contrast, learning with contamination is usually formulated as learning with nasty noise, where the goal is to achieve error $2\eta + \epsilon$ on the *target* distribution $(\mathcal{D}^*, f^*)$, where $f^* \in \mathcal{C}$ and $\eta$ is the noise rate. Therefore, given an algorithm for learning with contamination, one may obtain an agnostic learner with error guarantee $3\mathsf{opt} + \epsilon$, but it is not clear whether the guarantee of $\mathsf{opt} + \epsilon$ can be achieved in a black-box way. This is because in one formulation the error is measured with respect to the input distribution and in the other it is measured with respect to the clean target distribution.

Furthermore, many of the lower bounds for agnostic learning seem to exclusively rule out algorithms with error guarantees of $\mathsf{opt} + \epsilon$ [DKPZ21]. Therefore, these lower bounds do not transfer directly to learning with bounded contamination as defined in Definition 1.1. One natural way to go around this, is to observe that our result in Theorem 4.2 would also hold for the following stronger version of Definition 1.1.

**Definition E.1** (Agnostic BC-Learning). *An algorithm $\mathcal{A}$ is an agnostic BC-learner for $\mathcal{C} \subseteq \{\mathcal{X} \to \{\pm 1\}\}$ if on input $(\epsilon, \delta, \bar{S}_{\mathrm{inp}})$, where $\epsilon, \delta \in (0, 1)$, and $\bar{S}_{\mathrm{inp}}$ is generated by $\bar{\mathcal{D}}$ with bounded contamination $\eta$ for some labeled distribution $\bar{\mathcal{D}}$, and $\eta \in [0, 1)$, the algorithm $\mathcal{A}$ outputs some hypothesis $h : \mathcal{X} \to \{\pm 1\}$ such that with probability at least $1 - \delta$ over the clean examples in $\bar{S}_{\mathrm{inp}}$, and the randomness of $\mathcal{A}$:*

$$\mathbb{P}_{(\mathbf{x},y)\sim\bar{\mathcal{D}}}[y \neq h(\mathbf{x})] \leq 2\eta + \mathsf{opt}_{\mathrm{clean}} + \epsilon, \text{ where } \mathsf{opt}_{\mathrm{clean}} = \min_{f \in \mathcal{C}} \mathbb{P}_{(\mathbf{x},y)\sim\bar{\mathcal{D}}}[y \neq f(\mathbf{x})]$$

*The sample complexity of $\mathcal{A}$ is the minimum number of examples required to achieve the above guarantee. Moreover, a distribution-specific BC-learner with respect to some distribution $\mathcal{D}^*$ over $\mathcal{X}$ is a BC-learner that is guaranteed to work only when $\mathcal{D} = \mathcal{D}^*$.*

Note that Definition E.1 is a generalization of agnostic learning, since we can set $\eta = 0$. Our Theorem 4.2 holds even under this definition, and all the lower bounds for agnostic learning (Table 3)

are inherited as well. However, agnostic learning is known to be characterized by the $\mathcal{L}_1$-approximate degree [KKMS08, DSFT$^+$14, DKPZ21], whereas we require the stronger assumption of low $\mathcal{L}_2$-approximate degree. Therefore, while for most interesting classes our results are essentially tight, a complete characterization of efficient learnability in the setting of Definition E.1 remains open.

**Heavy Contamination.**   We give the following, information-theoretic lower bound for learning monotone functions with heavy contamination. Our lower bound highlights a separation between bounded and heavy contamination and justifies the need for a stronger notion of approximation than standard polynomial approximators.

**Theorem E.2** (Lower Bound for HC-Learning of Monotone Functions). *Let $\mathcal{C}$ be the class of monotone functions over $\{\pm 1\}^d$. Any HC-learner for $\mathcal{C}$ with respect to the uniform distribution over the hypercube $\mathrm{Unif}(\{\pm 1\}^d)$ requires sample complexity $2^{\Omega(d)}$, even when $\epsilon = 1/4$, $Q = 2$ and $\delta = 1/3$.*

*Proof of Theorem E.2.* Suppose that the ground truth is a constant function, i.e., either $f^* \equiv 1$ or $f^* \equiv -1$, and that $m \le 2^{d/10}/10$. Then, the adversary receives the $m$ clean examples $\bar{S}_{\mathrm{inp}}$ and draws $m$ additional i.i.d. examples $S'$ from $\mathrm{Unif}(\{\pm 1\}^d)$, labeling them according to $-f^*$. The input dataset is $\bar{S}_{\mathrm{inp}} = \bar{S}_{\mathrm{cln}} \cup \bar{S}'$.

Note, first, that no algorithm can achieve error better than $1/2$ with probability at least $1/2$, since guessing $f^*$ randomly is an optimal strategy because the distribution of the input dataset $\bar{S}_{\mathrm{inp}}$ is exactly the same in the two cases corresponding to $f^* \equiv -1$ and $f^* \equiv 1$. We will show that $Q \cdot \mathsf{opt}_{\mathrm{total}} = 0$ with high probability over the choice of $\bar{S}_{\mathrm{inp}}$.

Let $\mathbf{x}, \mathbf{x}'$ be two independent samples from $\mathrm{Unif}(\{\pm 1\}^d)$. We say that $\mathbf{x}$ and $\mathbf{x}'$ are comparable if $\mathbf{x} \le \mathbf{x}'$ or $\mathbf{x}' \le \mathbf{x}$, i.e., if $\mathbf{x}(i) \le \mathbf{x}'(i)$ for all $i$ or $\mathbf{x}'(i) \le \mathbf{x}(i)$ for all $i$. Since the coordinates are independent, we have

$$\mathbb{P}_{\mathbf{x},\mathbf{x}'}[\mathbf{x}(i) \le \mathbf{x}'(i)\,,\text{ for all } i \in [d]] = \prod_{i \in [d]} \mathbb{P}_{\mathbf{x},\mathbf{x}'}[\mathbf{x}(i) \le \mathbf{x}'(i)] = (3/4)^d$$

Similarly, for the other direction we have that $\mathbb{P}_{\mathbf{x},\mathbf{x}'}[\mathbf{x}(i) \ge \mathbf{x}'(i)\,,\text{ for all } i \in [d]] \le (3/4)^d$ and, overall, that $\mathbb{P}_{\mathbf{x},\mathbf{x}'}[\mathbf{x}, \mathbf{x}' \text{ are comparable}] \le 2 \cdot (3/4)^d$.

We have a set $\bar{S}_{\mathrm{inp}}$ of $2m$ examples. It suffices to show that no pair of examples in $S_{\mathrm{inp}}$ is comparable, because, then, any labeling of these examples is consistent with some monotone function and, hence, $\mathsf{opt}_{\mathrm{total}} = 0$. We overall have at most $4m^2$ pairs and each of them is comparable with probability at most $2 \cdot (3/4)^d$. Therefore, by a union bound, the probability that there is a pair that is comparable is at most $8m^2(3/4)^d \le 1/10$. □

# F   Omitted Proofs for Iterative Polynomial Filtering

We now give the proof of our main technical tool, Theorem 3.2. The proof follows the approach of [KSV24c], but has a number of technical differences. In particular, the algorithm of [KSV24c] only gives guarantees for non-negative polynomials and works with respect to the uniform distribution on the hypercube. Here, we preserve the expectation of any polynomial, with respect to any target hypercontractive distribution that can be sampled efficiently.

*Proof of Theorem 3.2.* We first observe that the family $\mathcal{P}$ of polynomials $p$ of degree at most $\ell$ for which $\mathbb{E}_{\mathbf{x} \sim S_{\mathrm{ref}}}[|p(\mathbf{x})|] \le \gamma$ and $\mathbb{E}_{\mathbf{x} \sim S_{\mathrm{ref}}}[(p(\mathbf{x}))^2] \le \beta$ can be described by $O(m_{\mathrm{ref}})$ linear constraints plus one convex quadratic constraint over the coefficient vectors whose dimension is $(d+1)^\ell$. Therefore, lines 3 and 5 can be implemented as convex programs in time $\mathrm{poly}(|S_{\mathrm{inp}}|, m_{\mathrm{ref}}, (d+1)^\ell)$. Lines 8 and 9 can be implemented with a single pass of the input points.

We first prove the following claim, which ensures that lines 8 and 9 are well defined.

**Claim.** *When $\lambda^* > \epsilon$, there exists $\tau^* \ge 0$ that satisfies the guarantees of line 8.*

*Proof.* Suppose, for contradiction, that for any $\tau \geq 0$ we have

$$\frac{|S|}{m} \mathop{\mathbb{P}}_{\mathbf{x} \sim S}[|p^*(\mathbf{x})| > \tau] < R \cdot \mathop{\mathbb{P}}_{\mathbf{x} \sim S_{\mathrm{ref}}}[|p^*(\mathbf{x})| > \tau] + \Delta$$

We may now integrate both sides of the inequality over $\tau \in [0, B]$, since it holds for all $\tau \geq 0$. Note that $|p^*(\mathbf{x})| \geq 0$ for all $\mathbf{x}$ and, therefore, the following are true

$$\mathop{\mathbb{E}}_{\mathbf{x} \sim S}[|p^*(\mathbf{x})|] = \int_{\tau=0}^{\infty} \mathop{\mathbb{P}}_{\mathbf{x} \sim S}[|p^*(\mathbf{x})| > \tau] \, d\tau = \int_{\tau=0}^{B} \mathop{\mathbb{P}}_{\mathbf{x} \sim S}[|p^*(\mathbf{x})| > \tau] \, d\tau \tag{F.1}$$

$$\mathop{\mathbb{E}}_{\mathbf{x} \sim S_{\mathrm{ref}}}[|p^*(\mathbf{x})|] = \int_{\tau=0}^{\infty} \mathop{\mathbb{P}}_{\mathbf{x} \sim S_{\mathrm{ref}}}[|p^*(\mathbf{x})| > \tau] \, d\tau \geq \int_{\tau=0}^{B} \mathop{\mathbb{P}}_{\mathbf{x} \sim S_{\mathrm{ref}}}[|p^*(\mathbf{x})| > \tau] \, d\tau \tag{F.2}$$

The second equality in Eq. (F.1) follows from the fact that for any $p^* \in \mathcal{P}$, $S$ contains only points such that $p^*(\mathbf{x}) \in [-B, B]$, due to line 3. The inequality in Eq. (F.2) follows from the fact that the integrated function is non-negative. Overall, we obtain the following inequality

$$\lambda^* = \frac{|S|}{m} \mathop{\mathbb{E}}_{\mathbf{x} \sim S}[p^*(\mathbf{x})] \leq \frac{|S|}{m} \mathop{\mathbb{E}}_{\mathbf{x} \sim S}[|p^*(\mathbf{x})|] < R \cdot \mathop{\mathbb{E}}_{\mathbf{x} \sim S_{\mathrm{ref}}}[|p^*(\mathbf{x})|] + \Delta B \leq \epsilon,$$

where the last inequality follows from the fact that $p^* \in \mathcal{P}$ and, hence, $\mathbb{E}_{\mathbf{x} \sim S_{\mathrm{ref}}}[|p^*(\mathbf{x})|] \leq \frac{\epsilon}{2R}$ and $\Delta = \frac{\epsilon}{2B}$. We have reached contradiction, because we showed that $\lambda^* \leq \epsilon$. $\qquad\square$

Note that due to the claim above, we have $\{\mathbf{x} \in S : |p^*(\mathbf{x})| > \tau^*\} \neq \emptyset$. This means that at each iteration we remove at least one point from the input dataset and, therefore, we do not need more than $M = |S_{\mathrm{inp}}|$ iterations. The following claim ensures that we only remove a small fraction of clean points from $S_{\mathrm{inp}}$.

**Claim.** *With probability at least $1 - \delta$ over $S_{\mathrm{cln}}$ and $S_{\mathrm{ref}}$, we have*

$$|(S_{\mathrm{cln}} \cap S_{\mathrm{inp}}) \setminus S_{\mathrm{filt}}| \leq \frac{\epsilon m}{2} + \frac{1}{R} \cdot |S_{\mathrm{inp}} \setminus S_{\mathrm{filt}}|$$

*Proof.* We will show that with high probability over the clean and reference datasets, the number of removed points from $S_{\mathrm{cln}}$ is small. First, we account for the initial filtering of line 3. In this step, we remove points $\mathbf{x}$ from $S_{\mathrm{inp}}$ that give large absolute values to polynomials in $\mathcal{P}$, i.e., polynomials who, in particular, have bounded second norms over $S_{\mathrm{ref}}$. Due to Lemma A.2 and since $m_{\mathrm{ref}} \geq (CAd)^{2\ell}(\log \frac{1}{\delta})^{4\ell+1}$, the probability that some $\mathbf{x}$ drawn from $\mathcal{D}$ gives $|p(\mathbf{x})| > B$ for some $p \in \mathcal{P}$ is at most $\epsilon/4$. Therefore, the total number of points removed from $S_{\mathrm{cln}}$ in this step follows the binomial distribution with $m$ number of trials and probability of success at most $\epsilon/4$. By a standard Chernoff bound, and since $m \geq \frac{C}{\epsilon} \log(1/\delta)$, with probability at least $1 - \delta/2$ we have

$$|\{\mathbf{x} \in S_{\mathrm{cln}} : |p(\mathbf{x})| > B \text{ for some } p \in \mathcal{P}\}| \leq \epsilon m/2 \tag{F.3}$$

It remains to account for the points removed in line 9. The removed points are always of the form $\{\mathbf{x} \in S : |p^*(\mathbf{x})| > \tau^*\}$ for some $p^* \in \mathcal{P}$. Moreover, according to line 8, we have

$$\frac{1}{m} \sum_{\mathbf{x} \in S} \mathbb{1}\{|p^*(\mathbf{x})| > \tau^*\} \geq R \cdot \mathop{\mathbb{P}}_{\mathbf{x} \sim S_{\mathrm{ref}}}[|p^*(\mathbf{x})| > \tau^*] + \Delta$$

Since $m, m_{\mathrm{ref}} \geq C' \frac{(d+1)^\ell + \log(1/\delta)}{(\Delta/R)^2}$ for some sufficiently large universal constant $C' \geq 1$ and the VC dimension of the class of polynomial threshold functions of degree $\ell$ is at most $(d+1)^\ell$, we have that with probability at least $1 - \delta/2$ over $S_{\mathrm{cln}}, S_{\mathrm{ref}}$ the following holds for all polynomials $p$ of degree at most $\ell$ and all $\tau \geq 0$

$$R \cdot \mathop{\mathbb{P}}_{\mathbf{x} \sim S_{\mathrm{cln}}}[|p(\mathbf{x})| > \tau] \leq R \cdot \mathop{\mathbb{P}}_{\mathbf{x} \sim S_{\mathrm{ref}}}[|p(\mathbf{x})| > \tau] + \Delta,$$

since $S_{\mathrm{cln}}$ and $S_{\mathrm{ref}}$ are both i.i.d. samples from the distribution $\mathcal{D}^*$ and $\mathbb{1}\{|p(\mathbf{x})| > \tau\}$ is equal to the sum of two degree-$\ell$ polynomial threshold functions, $\mathbb{1}\{p(\mathbf{x}) > \tau\} + \mathbb{1}\{p(\mathbf{x}) < -\tau\}$. Therefore:

$$|\{\mathbf{x} \in S : |p^*(\mathbf{x})| > \tau^*\}| \geq R \cdot |\{\mathbf{x} \in S_{\mathrm{cln}} : |p^*(\mathbf{x})| > \tau^*\}|,$$

for any update we make. Summing over all the updates, we obtain that the total number of points removed from $S_{\mathrm{cln}}$ by the updates of line 9 is at most

$$(1/R) \cdot |S_{\mathrm{inp}} \setminus S_{\mathrm{filt}}| \tag{F.4}$$

By combining Eq. (F.3) and (F.4), we obtain the desired result, where the probability of failure is at most $\delta$ due to a union bound. $\qquad\square$

So far, we have proven the first part of Theorem 3.2. We will now prove the second part, by showing the following claim.

**Claim.** *Any polynomial $p$ of degree at most $\ell$ and $\mathbb{E}_{\mathbf{x}\sim\mathcal{D}}[|p(\mathbf{x})|] \leq \epsilon/(4R)$, lies within $\mathcal{P}$ with probability at least $1 - \delta$ over $S_{\mathrm{ref}}$.*

*Proof.* Let $\mathcal{D} = \mathcal{D}^*$. Fix any polynomial $p$ with $\mathbb{E}_{\mathbf{x}\sim\mathcal{D}}[|p(\mathbf{x})|] \leq \epsilon/(4R)$. Due to the hypercontractivity of $\mathcal{D}$ (Definition 3.1), we have $\mathbb{E}_{\mathbf{x}\sim\mathcal{D}}[(p(\mathbf{x}))^2] \leq (2A)^{2\ell} = \beta/2$, since $\mathbb{E}_{\mathbf{x}\sim\mathcal{D}}[|p(\mathbf{x})|] \leq 1$. By Lemma A.1, since $m_{\mathrm{ref}}$ is large enough, w.p. at least $1 - \frac{\delta}{2}$ over $S_{\mathrm{ref}}$, we have $\mathbb{E}_{\mathbf{x}\sim S_{\mathrm{ref}}}[(p(\mathbf{x}))^2] \leq \beta$.

It remains to show that with probability at least $1-\delta/2$, we have $\mathbb{E}_{\mathbf{x}\sim S_{\mathrm{ref}}}[|p(\mathbf{x})|] \leq \gamma = \epsilon/(2R)$. The following bound can be obtained for any $t \geq 1$ by applying the Marcinkiewicz–Zygmund inequality (see [Fer14]), due to the fact that $S_{\mathrm{ref}}$ consists of $m_{\mathrm{ref}}$ i.i.d. examples.

$$\mathbb{P}_{S_{\mathrm{ref}}\sim\mathcal{D}^{m_{\mathrm{ref}}}}\left[\left|\mathbb{E}_{\mathbf{x}\sim S_{\mathrm{ref}}}[|p(\mathbf{x})|] - \mathbb{E}_{\mathbf{x}'\sim\mathcal{D}}[|p(\mathbf{x}')|]\right| > \frac{\epsilon}{4R}\right] \leq 2\left(\frac{32R^2 t}{\epsilon^2 m_{\mathrm{ref}}}\right)^t \mathbb{E}_{\mathbf{x}\sim\mathcal{D}}\left[\left||p(\mathbf{x})| - \mathbb{E}_{\mathbf{x}'\sim\mathcal{D}}[|p(\mathbf{x}')|]\right|^{2t}\right]$$

To conclude the proof of the claim, we use the simple inequality $(a + b)^{2t} \leq 4^t \max\{a^{2t}, b^{2t}\}$ for all $a, b, t \geq 0$ as well as hypercontractivity to bound the expectation on the right-hand side by $(8At)^{2\ell t}\mathbb{E}_{\mathbf{x}\sim\mathcal{D}}[|p(\mathbf{x})|]^{2t}$. Due to the choice of $m_{\mathrm{ref}}$, if we choose $t = \log(1/\delta)$, we obtain a bound of $\delta/2$, as desired. $\qquad\square$

This concludes the proof of Theorem 3.2. $\qquad\square$

## G   Omitted Details for Learning with Contamination

### G.1   Bounded Contamination

We give the proof of our main result on learning with bounded contamination (Theorem 4.2) below.

*Proof of Theorem 4.2.* The algorithm receives a dataset $\bar{S}_{\mathrm{inp}}$ generated by $(\mathcal{D}^*, f^*)$ with bounded contamination of rate $\eta \in (0, 1)$, where $f^* \in \mathcal{C}$ and $\eta, f^*$ are unknown, draws a set $S_{\mathrm{ref}}$ of $m_{\mathrm{ref}} = \frac{(C' A d)^{4\ell}}{\epsilon^6}(\log\frac{1}{\delta})^{8\ell+1}$ i.i.d. unlabeled examples from $\mathcal{D}^*$, where $C' \geq 1$ is a sufficiently large universal constant and does the following.

1. First, the algorithm runs the filtering procedure of Theorem 3.2 (that is, Algorithm 1) on input $(S_{\mathrm{inp}}, S_{\mathrm{ref}}, m = |S_{\mathrm{inp}}|, 2\ell, R = 2, 24\epsilon^2/\sqrt{C})$ to form the filtered dataset $\bar{S}_{\mathrm{filt}}$, where the labels are consistent with $\bar{S}_{\mathrm{inp}}$.

2. Then, the algorithm finds a polynomial $\hat{p}$ of degree at most $\ell$ that minimizes the following convex objective.

$$\hat{p} = \arg\min_{p} \mathbb{E}_{(\mathbf{x},y)\sim\bar{S}_{\mathrm{filt}}}[|y - p(\mathbf{x})|]$$

$$\text{s.t. } p \text{ has degree at most } \ell$$

3. The algorithm outputs $h(\mathbf{x}) = \mathrm{sign}(\hat{p}(\mathbf{x}) + \hat{\tau})$, where $\hat{\tau} \in \mathbb{R}$ minimizes the one-dimensional objective $\mathbb{P}_{(\mathbf{x},y)\sim\bar{S}_{\mathrm{filt}}}[y \neq \mathrm{sign}(\hat{p}(\mathbf{x}) + \tau)]$ over $\tau \in \mathbb{R}$.

We will now bound the error of $h$ on the clean dataset $\bar{S}_{\mathrm{cln}}$ according to which $\bar{S}_{\mathrm{inp}}$ was formed (see Definition 1.1). This suffices because, due to standard VC theory, and since $h$ is a polynomial threshold function of degree at most $\ell$, as long as $m = |\bar{S}_{\mathrm{cln}}| \geq C'\frac{(d+1)^\ell + \log\frac{1}{\delta}}{\epsilon^2}$ for some sufficiently large universal constant $C' \geq 1$, we have that $\mathbb{P}_{\mathbf{x}\sim\mathcal{D}}[f^*(\mathbf{x}) \neq h(\mathbf{x})]$ is approximately equal to its empirical counterpart $\mathbb{P}_{(\mathbf{x},y)\sim\bar{S}_{\mathrm{inp}}}[y \neq h(\mathbf{x})]$.

Consider $p^*$ to be a polynomial of degree at most $\ell$ such that $\mathbb{E}_{\mathbf{x}\sim\mathcal{D}}[(f^*(\mathbf{x}) - p^*(\mathbf{x}))^2] \leq \epsilon'$, where $\epsilon' = \epsilon^4/C$. Since $p^*$ approximates a Boolean-valued function $f^*$, the values of $p^*$ should, in

expectation, be close to either $1$ or $-1$. In particular, we obtain the following bound using the Cauchy-Schwarz inequality:

$$
\begin{aligned}
\mathbb{E}_{\mathbf{x}\sim\mathcal{D}^*}[|(p^*(\mathbf{x}))^2 - 1|] &= \mathbb{E}_{\mathbf{x}\sim\mathcal{D}}[|(p^*(\mathbf{x}))^2 - (f^*(\mathbf{x}))^2|] \\
&= \mathbb{E}_{\mathbf{x}\sim\mathcal{D}^*}[|p^*(\mathbf{x}) - f^*(\mathbf{x})| \cdot |p^*(\mathbf{x}) + f^*(\mathbf{x})|] \\
&\leq \sqrt{\mathbb{E}_{\mathbf{x}\sim\mathcal{D}^*}[(p^*(\mathbf{x}) - f^*(\mathbf{x}))^2]} \cdot \sqrt{\mathbb{E}_{\mathbf{x}\sim\mathcal{D}^*}[(p^*(\mathbf{x}) + f^*(\mathbf{x}))^2]} \\
&\leq \epsilon' + \sqrt{2\epsilon'} \leq 3\epsilon^2/\sqrt{C}
\end{aligned}
\tag{G.1}
$$

Note that $q(\mathbf{x}) = (p^*(\mathbf{x}))^2 - 1$ is a polynomial of degree at most $2\ell$. Therefore, Algorithm 1 can be used to filter out the points $\mathbf{x}$ such that $q(\mathbf{x})$ is too large and, in particular, due to Theorem 3.2, the following holds with probability at least $1 - \delta/4$ over the random choice of $S_{\text{ref}}$ and $\bar{S}_{\text{cln}}$ as long as $m \geq C'\frac{(2A(d+1))^{4\ell}}{\epsilon^6}\log\frac{1}{\delta}$ and $m_{\text{ref}} \geq \frac{(C'Ad)^{4\ell}}{\epsilon^6}(\log\frac{1}{\delta})^{8\ell+1}$:

$$
\sum_{\mathbf{x}\in S_{\text{filt}}} q(\mathbf{x}) \leq \frac{\epsilon^2 m}{C''} , \tag{G.2}
$$

where $C'' = \sqrt{C}/24$. We will now show that this bound is sufficient for our purposes.

Consider the quantity $P = \frac{|\bar{S}_{\text{filt}}|}{m}\mathbb{P}_{(\mathbf{x},y)\sim\bar{S}_{\text{filt}}}[y \neq h(\mathbf{x})]$. We first bound the quantity $P$ by $P \leq \frac{|\bar{S}_{\text{filt}}|}{2m}\mathbb{E}_{(\mathbf{x},y)\sim\bar{S}_{\text{filt}}}[|y - \widehat{p}(\mathbf{x})|]$, where the factor $1/2$ appears due to the fact that the choice of $\widehat{\tau}$ is optimal (and a random choice would yield the factor $1/2$ with positive probability, see [KKMS08]). Furthermore, we have the following, due to the fact that $\widehat{p}$ is optimal among low-degree polynomials for $\bar{S}_{\text{filt}}$ with respect to the absolute error.

$$
\begin{aligned}
P &\leq \frac{1}{2m}\sum_{(\mathbf{x},y)\in\bar{S}_{\text{filt}}}|y - \widehat{p}(\mathbf{x})| \leq \frac{1}{2m}\sum_{(\mathbf{x},y)\in\bar{S}_{\text{filt}}}|y - p^*(\mathbf{x})| \\
&\leq \frac{1}{2m}\sum_{(\mathbf{x},y)\in\bar{S}_{\text{filt}}\cap\bar{S}_{\text{cln}}}|f^*(\mathbf{x}) - p^*(\mathbf{x})| + \frac{1}{2m}\sum_{(\mathbf{x},y)\in\bar{S}_{\text{filt}}\setminus\bar{S}_{\text{cln}}}|y - p^*(\mathbf{x})| \\
&\leq \frac{1}{2m}\sum_{(\mathbf{x},y)\in\bar{S}_{\text{cln}}}|f^*(\mathbf{x}) - p^*(\mathbf{x})| + \frac{|\bar{S}_{\text{filt}}\setminus\bar{S}_{\text{cln}}|}{2m} + \frac{1}{2m}\sum_{(\mathbf{x},y)\in\bar{S}_{\text{filt}}\setminus\bar{S}_{\text{cln}}}|p^*(\mathbf{x})|
\end{aligned}
\tag{G.3}
$$

For the first term, observe that we choose $m = |\bar{S}_{\text{inp}}| = |\bar{S}_{\text{cln}}|$ and therefore we have $\frac{1}{2m}\sum_{(\mathbf{x},y)\in\bar{S}_{\text{cln}}}|f^*(\mathbf{x}) - p^*(\mathbf{x})| = \frac{1}{2}\mathbb{E}_{\mathbf{x}\sim S_{\text{cln}}}[|f^*(\mathbf{x}) - p^*(\mathbf{x})|]$. In order to bound this quantity, we use the fact that $S_{\text{cln}}$ is a set of $m$ i.i.d. samples from a hypercontractive distribution, combining the Marcinkiewicz-Zygmund inequality (which is a generalization of Chebyshev's inequality to higher moments), hypercontractivity and boundedness of $f^*$ to obtain that, as long as $m \geq \frac{1}{\epsilon^2}(C'A)^{2\ell}(\log(1/\delta))^{2\ell+1}$, with probability at least $1 - \delta/4$ over the choice of $S_{\text{cln}}$ we have:

$$
\begin{aligned}
\mathbb{E}_{\mathbf{x}\sim S_{\text{cln}}}[|f^*(\mathbf{x}) - p^*(\mathbf{x})|] &\leq \mathbb{E}_{\mathbf{x}\sim\mathcal{D}^*}[|f^*(\mathbf{x}) - p^*(\mathbf{x})|] + \epsilon/6 \\
&\leq \sqrt{\mathbb{E}_{\mathbf{x}\sim\mathcal{D}^*}[(f^*(\mathbf{x}) - p^*(\mathbf{x}))^2]} + \epsilon/6 \\
&\leq \epsilon^2/\sqrt{C} + \epsilon/6 \leq \epsilon/3
\end{aligned}
\tag{G.4}
$$

For the last term in Eq. (G.3), we first use the basic fact that the variance of any random variable is non-negative and hence $\mathbb{E}_{\mathbf{x}\sim S_{\text{filt}}\setminus S_{\text{cln}}}[|p^*(\mathbf{x})|] \leq (\mathbb{E}_{\mathbf{x}\sim S_{\text{filt}}\setminus S_{\text{cln}}}[(p^*(\mathbf{x}))^2])^{1/2}$. Then, we observe that $\mathbb{E}_{\mathbf{x}\sim S_{\text{filt}}\setminus S_{\text{cln}}}[(p^*(\mathbf{x}))^2] = \mathbb{E}_{\mathbf{x}\sim S_{\text{filt}}\setminus S_{\text{cln}}}[q(\mathbf{x})] + 1$. Overall, we have:

$$
\frac{1}{2m}\sum_{\mathbf{x}\in S_{\text{filt}}\setminus S_{\text{cln}}}|p^*(\mathbf{x})| \leq \frac{|S_{\text{filt}}\setminus S_{\text{cln}}|}{2m}\left(1 + \mathbb{E}_{\mathbf{x}\sim S_{\text{filt}}\setminus S_{\text{cln}}}[q(\mathbf{x})]\right)^{1/2} \tag{G.5}
$$

We will now bound the quantity $P' = \mathbb{E}_{\mathbf{x} \sim S_{\text{filt}} \setminus S_{\text{cln}}}[q(\mathbf{x})] = \frac{1}{|S_{\text{filt}} \setminus S_{\text{cln}}|} \sum_{\mathbf{x} \in S_{\text{filt}} \setminus S_{\text{cln}}} q(\mathbf{x})$.

$$P' = \frac{1}{|S_{\text{filt}} \setminus S_{\text{cln}}|} \sum_{\mathbf{x} \in S_{\text{filt}}} q(\mathbf{x}) - \frac{1}{|S_{\text{filt}} \setminus S_{\text{cln}}|} \sum_{\mathbf{x} \in S_{\text{filt}} \cap S_{\text{cln}}} q(\mathbf{x})$$

$$\overset{\text{(G.2)}}{\leq} \frac{\epsilon^2 m}{C'' |S_{\text{filt}} \setminus S_{\text{cln}}|} + \frac{1}{|S_{\text{filt}} \setminus S_{\text{cln}}|} \sum_{\mathbf{x} \in S_{\text{filt}} \cap S_{\text{cln}}} |q(\mathbf{x})|$$

$$\leq \frac{\epsilon^2 m}{C'' |S_{\text{filt}} \setminus S_{\text{cln}}|} + \frac{m}{|S_{\text{filt}} \setminus S_{\text{cln}}|} \underset{\mathbf{x} \sim S_{\text{cln}}}{\mathbb{E}}[|q(\mathbf{x})|] \tag{G.6}$$

The first inequality in the expression above follows from the guarantee of Eq. (G.2), which is provided by the filtering algorithm. We may now use the Marcinkiewicz-Zygmund inequality once more to obtain that, as long as $m \geq \frac{1}{\epsilon^4}(C'A)^{4\ell}(\log(1/\delta))^{4\ell+1}$, with probability at least $1 - \delta/4$, we have:

$$\underset{\mathbf{x} \sim S_{\text{cln}}}{\mathbb{E}}[|q(\mathbf{x})|] \leq \underset{\mathbf{x} \sim \mathcal{D}^*}{\mathbb{E}}[|q(\mathbf{x})|] + \epsilon^2/\sqrt{C} \leq \epsilon^2/C'',$$

where the last inequality follows from Eq. (G.1), the fact that $q(\mathbf{x}) = (p^*(\mathbf{x}))^2 - 1$, and recall that $C'' = \sqrt{C}$. Overall, we obtain the bound $P' \leq \frac{2\epsilon^2 m}{C''|S_{\text{filt}} \setminus S_{\text{cln}}|}$. Note that since for any $a \geq 0$ we have $\sqrt{1+a} \leq 1 + \sqrt{a}$, by substituting the bound for $P'$ in Eq. (G.5), we obtain

$$\frac{1}{2m} \sum_{\mathbf{x} \in S_{\text{filt}} \setminus S_{\text{cln}}} |p^*(\mathbf{x})| \leq \frac{|S_{\text{filt}} \setminus S_{\text{cln}}|}{2m} + \frac{\epsilon}{\sqrt{C''/2}},$$

where we used the fact that $|S_{\text{filt}} \setminus S_{\text{cln}}| \leq m$ to bound the factor $\sqrt{|S_{\text{filt}} \setminus S_{\text{cln}}|/m}$ in the second term of the above bound by 1. In total, by choosing $C$ appropriately large, we have the following bound, where we combine Eq. (G.3), (G.4), and (G.5).

$$P = \frac{1}{m} \sum_{(\mathbf{x},y) \in \bar{S}_{\text{filt}}} \mathbb{1}\{y \neq h(\mathbf{x})\} \leq \frac{|\bar{S}_{\text{filt}} \setminus \bar{S}_{\text{cln}}|}{m} + \frac{2\epsilon}{3},$$

with probability at least $1 - 3\delta/4$ over the random choice of $\bar{S}_{\text{inp}}$ and $S_{\text{ref}}$.

We will now bound $\mathbb{P}_{(\mathbf{x},y) \sim \bar{S}_{\text{inp}}}[y \neq h(\mathbf{x})]$ by $2\eta + 5\epsilon/6$. We have the following:

$$\underset{(\mathbf{x},y) \sim \bar{S}_{\text{cln}}}{\mathbb{P}}[y \neq h(\mathbf{x})] = \frac{1}{m} \sum_{(\mathbf{x},y) \in \bar{S}_{\text{cln}}} \mathbb{1}\{y \neq h(\mathbf{x})\}$$

$$= \frac{1}{m} \sum_{(\mathbf{x},y) \in \bar{S}_{\text{cln}} \cap \bar{S}_{\text{filt}}} \mathbb{1}\{y \neq h(\mathbf{x})\} + \frac{1}{m} \sum_{(\mathbf{x},y) \in \bar{S}_{\text{cln}} \setminus \bar{S}_{\text{filt}}} \mathbb{1}\{y \neq h(\mathbf{x})\}$$

$$\leq \sum_{(\mathbf{x},y) \in \bar{S}_{\text{filt}}} \mathbb{1}\{y \neq h(\mathbf{x})\} + \frac{|\bar{S}_{\text{cln}} \setminus \bar{S}_{\text{filt}}|}{m}$$

$$\leq \frac{|\bar{S}_{\text{filt}} \setminus \bar{S}_{\text{cln}}| + |(\bar{S}_{\text{cln}} \cap \bar{S}_{\text{inp}}) \setminus \bar{S}_{\text{filt}}| + |\bar{S}_{\text{cln}} \setminus \bar{S}_{\text{inp}}|}{m} + 2\epsilon/3$$

In order to complete the proof, we use the guarantee of Algorithm 1 that with probability at least $1 - \delta/4$, we have

$$|(\bar{S}_{\text{cln}} \cap \bar{S}_{\text{inp}}) \setminus \bar{S}_{\text{filt}}| \leq \frac{1}{2}|\bar{S}_{\text{inp}} \setminus \bar{S}_{\text{filt}}| + \frac{\epsilon m}{12}$$

$$= \frac{1}{2}|(\bar{S}_{\text{cln}} \cap \bar{S}_{\text{inp}}) \setminus \bar{S}_{\text{filt}}| + \frac{1}{2}|(\bar{S}_{\text{inp}} \setminus \bar{S}_{\text{cln}}) \setminus \bar{S}_{\text{filt}}| + \epsilon m/12.$$

Therefore, we have $|(\bar{S}_{\text{cln}} \cap \bar{S}_{\text{inp}}) \setminus \bar{S}_{\text{filt}}| \leq |(\bar{S}_{\text{inp}} \setminus \bar{S}_{\text{cln}}) \setminus \bar{S}_{\text{filt}}| + \epsilon m/6$. Note that the union of the sets $(\bar{S}_{\text{inp}} \setminus \bar{S}_{\text{cln}}) \setminus \bar{S}_{\text{filt}}$ and $\bar{S}_{\text{filt}} \setminus \bar{S}_{\text{cln}}$ is disjoint and equals to the set of adversarial examples $\bar{S}_{\text{adv}} = \bar{S}_{\text{inp}} \setminus \bar{S}_{\text{cln}}$. By the definition of the noise model (Definition 1.1), we have that $|\bar{S}_{\text{adv}}| \leq \eta m$ and $|\bar{S}_{\text{cln}} \setminus \bar{S}_{\text{inp}}| = |\bar{S}_{\text{adv}}| \leq \eta m$. Therefore, we overall obtain the desired bound of

$$\underset{(\mathbf{x},y) \sim \bar{S}_{\text{cln}}}{\mathbb{P}}[y \neq h(\mathbf{x})] \leq 2\eta + \frac{5\epsilon}{6}.$$

This concludes the proof of Theorem 4.2. $\qquad\square$

## G.2 Heavy Contamination

In Table 4, we provide a number of end-to-end results for learning with heavy contamination, which are obtained by combining Theorem 4.4 with appropriate known bounds on the sandwiching degree (see Appendix C.2). The first row of Table 4 gives the bound $d^{\tilde{O}(Q^2 k^6/\epsilon^4)}$ for learning intersections of $k$ halfspaces with $Q$-heavy contamination by choosing $s = t = k$, as well as the bound $d^{\tilde{O}(Q^2 k^4 4^k/\epsilon^4)}$ for arbitrary functions of $k$ halfspaces, with the choice $s = 2^k$, $t = k$.

| Concept Class | Target Marginal | Runtime |
|:---:|:---:|:---:|
| Depth-$t$, Size-$s$ Decision Trees of Halfspaces | $\mathcal{N}_d$ or $\mathrm{Unif}_d$ | $d^{\tilde{O}(Q^2 t^4 s^2/\epsilon^4)}$ |
| Depth-$t$, Size-$s$ Boolean Circuits | $\mathrm{Unif}_d$ | $d^{O(\log s)^{O(t)} \log(Q/\epsilon)}$ |
| Degree-$k$ PTFs | $\mathcal{N}_d$ | $d^{\tilde{O}_k((Q/\epsilon^2)^{4k \cdot 7^k})}$ |
| Degree-2 PTFs | $\mathrm{Unif}_d$ | $d^{\tilde{O}(Q^9/\epsilon^{18})}$ |

Table 4: Bounds on the time complexity of learning with $Q$-heavy contamination up to error $Q \cdot \mathsf{opt}_{\mathrm{total}} + \epsilon$ and failure probability $\delta = 0.01$.

We now give the proof of our main result on learning with heavy contamination (Theorem 4.4).

*Proof of Theorem 4.4.* The algorithm receives a $Q$-heavily contaminated dataset $\bar{S}_{\mathrm{inp}}$, draws a set $S_{\mathrm{ref}}$ of $m_{\mathrm{ref}} = \frac{Q^2 (C'Ad)^{2\ell}}{\epsilon^5} (\log \frac{1}{\delta})^{4\ell+1}$ i.i.d. unlabeled examples from $\mathcal{D} = \mathcal{D}^*$, where $C' \geq 1$ is a sufficiently large universal constant and does the following.

1. First, the algorithm runs the filtering procedure of Theorem 3.2 (that is, Algorithm 1) on input $(S_{\mathrm{inp}}, S_{\mathrm{ref}}, m = \frac{Q^2 (C'Ad)^{2\ell} \log \frac{1}{\delta}}{\epsilon^5}, \ell, R = \frac{2Q}{\epsilon}, \epsilon/3)$ to form the filtered dataset $\bar{S}_{\mathrm{filt}}$.

2. Then, the algorithm finds a polynomial $\widehat{p}$ of degree at most $\ell$ that minimizes the following convex objective.
$$\widehat{p} = \arg\min_{p} \ \mathbb{E}_{(\mathbf{x},y)\sim\bar{S}_{\mathrm{filt}}} [|y - p(\mathbf{x})|]$$
$$\text{s.t. } p \text{ has degree at most } \ell$$

3. The algorithm outputs $h(\mathbf{x}) = \mathrm{sign}(\widehat{p}(\mathbf{x}) + \widehat{\tau})$, where $\widehat{\tau} \in \mathbb{R}$ minimizes the one-dimensional objective $\mathbb{P}_{(\mathbf{x},y)\sim\bar{S}_{\mathrm{filt}}}[y \neq \mathrm{sign}(\widehat{p}(\mathbf{x}) + \tau)]$ over $\tau \in \mathbb{R}$.

We will now bound the error of $h$ on the clean distribution $\bar{\mathcal{D}}$. To this end, we first note that the empirical error of $h$ on the clean samples $\bar{S}_{\mathrm{cln}}$ is, with high probability, close to the error on $\bar{\mathcal{D}}$, since $|\bar{S}_{\mathrm{cln}}|$ is large enough and $h$ is a polynomial threshold function of degree at most $\ell$. It therefore, suffices to bound the error of $h$ on $\bar{S}_{\mathrm{cln}}$. We have the following.

$$\mathbb{P}_{(\mathbf{x},y)\sim\bar{S}_{\mathrm{cln}}}[y \neq h(\mathbf{x})] = \frac{1}{m} \sum_{(\mathbf{x},y)\sim\bar{S}_{\mathrm{cln}}\cap\bar{S}_{\mathrm{filt}}} \mathbb{1}\{y \neq h(\mathbf{x})\} + \frac{1}{m} \sum_{(\mathbf{x},y)\sim\bar{S}_{\mathrm{cln}}\setminus\bar{S}_{\mathrm{filt}}} \mathbb{1}\{y \neq h(\mathbf{x})\}$$
$$\leq \frac{1}{m} \sum_{(\mathbf{x},y)\sim\bar{S}_{\mathrm{filt}}} \mathbb{1}\{y \neq h(\mathbf{x})\} + \frac{1}{m} \cdot |\bar{S}_{\mathrm{cln}} \setminus \bar{S}_{\mathrm{filt}}| \tag{G.7}$$

The second term in the bound above can be bounded by $\epsilon/3$, due to part 1 of Theorem 3.2. It remains to bound the first term.

Consider $f^* \in \mathcal{C}$ to be the concept in class $\mathcal{C}$ with minimum error on $\bar{S}_{\mathrm{filt}}$, i.e., we have that $f^* = \arg\min_{f\in\mathcal{C}} \mathbb{P}_{(\mathbf{x},y)\in\bar{S}_{\mathrm{filt}}}[y \neq f(\mathbf{x})]$. Note that there is some $f' \in \mathcal{C}$ that makes at most $M \cdot \mathsf{opt}_{\mathrm{total}}$ errors on the whole input dataset $\bar{S}_{\mathrm{inp}}$, due to the definition of $\mathsf{opt}_{\mathrm{total}}$ (see Definition 1.4). We have that $\mathbb{P}_{(\mathbf{x},y)\in\bar{S}_{\mathrm{filt}}}[y \neq f^*(\mathbf{x})] \leq \mathbb{P}_{(\mathbf{x},y)\in\bar{S}_{\mathrm{filt}}}[y \neq f'(\mathbf{x})] \leq M \cdot \mathsf{opt}_{\mathrm{total}}/|\bar{S}_{\mathrm{filt}}|$.

Moreover, let $p_{\text{up}}, p_{\text{down}}$ be the $\frac{\epsilon^2}{72Q}$-sandwiching polynomials for $f^*$ and $p = p_{\text{up}} - p_{\text{down}}$, which is a non-negative polynomial with $\mathbb{E}_{\mathbf{x} \sim \mathcal{D}}[|p(\mathbf{x})|] \leq \frac{\epsilon^2}{72Q}$. Due to part 2 of Theorem 3.2, the expectation of $p$ under $S_{\text{filt}}$ is at most $\frac{\epsilon m}{3|\bar{S}_{\text{filt}}|}$, i.e., $\mathbb{E}_{\mathbf{x} \sim S_{\text{filt}}}[p_{\text{up}}(\mathbf{x}) - p_{\text{down}}(\mathbf{x})] \leq \frac{\epsilon m}{3|\bar{S}_{\text{filt}}|}$. Since $p_{\text{up}} \geq f^* \geq p_{\text{down}}$, we also have that

$$\mathbb{E}_{\mathbf{x} \sim S_{\text{filt}}}[|f^*(\mathbf{x}) - p_{\text{down}}(\mathbf{x})|] \leq \frac{\epsilon m}{3|\bar{S}_{\text{filt}}|}$$

We now observe that $h$ is the output of the $\mathcal{L}_1$ polynomial regression algorithm on $\bar{S}_{\text{filt}}$ and, therefore, according to Theorem A.3 by [KKMS08], we overall have the following bound

$$\mathbb{P}_{(\mathbf{x},y) \sim \bar{S}_{\text{filt}}}[y \neq h(\mathbf{x})] \leq \frac{M}{|\bar{S}_{\text{filt}}|} \cdot \mathsf{opt}_{\text{total}} + \frac{\epsilon m}{3|\bar{S}_{\text{filt}}|} \, ,$$

which implies the desired bound:

$$\frac{1}{m} \sum_{(\mathbf{x},y) \sim \bar{S}_{\text{filt}}} \mathbb{1}\{y \neq h(\mathbf{x})\} \leq \frac{M}{m} \cdot \mathsf{opt}_{\text{total}} + \frac{\epsilon m}{2} \leq Q \cdot \mathsf{opt}_{\text{total}} + \epsilon/3 \tag{G.8}$$

The result follows by combining Eq. (G.7) and (G.8). $\qquad\square$

*Remark* G.1. Our result on HC-learning highlights the following analogy:

- Learning with heavy contamination can be thought of as a version of testable agnostic learning (Definition H.1), where the algorithm is asked to find a subset of the input that is structured enough, instead of merely verifying whether the whole input is structured.

- Similarly, in the context of learning with distribution shift, PQ learning [GKKM20] requires finding a subset of an unlabeled test dataset where the learner is confident in its own predictions, while TDS learning [KSV24b] aims to verify whether the whole unlabeled test dataset is drawn from some distribution that is similar to the one the learner has trained on.

- In [GSSV24], it is shown that by using a spectral iterative filtering algorithm, TDS learning results can be extended to PQ learning. Here, we complete the analogy by showing that our iterative filtering algorithm can extend known results from testable learning (i.e., that sandwiching is sufficient, see [GKK23]) to HC-learning.

*Remark* G.2. One could define a hybrid model where a bounded fraction of clean points are replaced adversarially and a proportionally large number of adversarial points are then added. Our results should apply to this setting as well, but we focus on BC and HC learning separately for simplicity of presentation.

# H   Applications to Tolerant Testable Learning

We give new results for testable learning [RV23] and tolerant testable learning [GSSV24].

**Definition H.1** (Tolerant Testable Learning [GSSV24])**.** *An algorithm $\mathcal{A}$ is a tolerant tester-learner for $\mathcal{C} \subseteq \{\mathcal{X} \to \{\pm 1\}\}$ with respect to some target distribution $\mathcal{D}^*$ over $\mathcal{X}$ if on input $(\epsilon, \delta, \tau, \bar{S}_{\text{inp}})$, where $\epsilon, \tau, \delta \in (0,1)$ and $\bar{S}_{\text{inp}}$ is a set of i.i.d. examples from some arbitrary distribution $\bar{\mathcal{D}}$, the algorithm $\mathcal{A}$ outputs either outputs $\mathrm{Reject}$ or outputs $(\mathrm{Accept}, h)$, where $h : \mathcal{X} \to \{\pm 1\}$ such that with probability at least $1 - \delta$ over $\bar{S}_{\text{inp}}$, and the randomness of $\mathcal{A}$, the following conditions hold.*

1. *(Soundness) Upon acceptance, the error of $h$ is bounded as follows:*

$$\mathbb{P}_{(\mathbf{x},y) \sim \bar{\mathcal{D}}}[y \neq h(\mathbf{x})] \leq \min_{f \in \mathcal{C}} \mathbb{P}_{(\mathbf{x},y) \sim \bar{\mathcal{D}}}[y \neq f(\mathbf{x})] + \tau + \epsilon$$

2. *(Completeness) If $\mathrm{d}_{\text{TV}}(\mathcal{D}, \mathcal{D}^*) \leq \tau$, where $\mathcal{D}$ is the marginal of $\bar{\mathcal{D}}$ on $\mathcal{X}$, then $\mathcal{A}$ accepts.*

*The sample complexity of $\mathcal{A}$ is the minimum number of examples required to achieve the above guarantee.*

Previous work by [GSSV24] showed that the existence of $\mathcal{L}_2$ sandwiching polynomials with bounded coefficients, i.e., polynomials $p_{\text{up}}, p_{\text{down}}$ such that $\mathbb{E}_{\mathbf{x} \sim \mathcal{D}^*}[(p_{\text{up}}(\mathbf{x}) - p_{\text{down}}(\mathbf{x}))^2] \leq \epsilon$ implies tolerant testable learning. Here, we relax this condition to $\mathcal{L}_1$ sandwiching polynomials (Definition 4.3), without requiring any bound on the coefficients. Note that even for non-tolerant testable learning, where $\mathcal{L}_1$ sandwiching is known to suffice [GKK23], all prior work required bounds on the coefficients.

**Theorem H.2** (Sandwiching implies Tolerant Testable Learning). *Let $\epsilon, \delta, \tau \in (0,1)$ and $Q \geq 1$. Let $\mathcal{D}^*$ be some $A$-hypercontractive distribution over a $d$-dimensional space and let $\mathcal{C}$ be a concept class whose $\frac{\epsilon}{C(1+\tau/\epsilon)}$-sandwiching degree w.r.t. $\mathcal{D}^*$ is $\ell$ for some large enough constant $C \geq 1$. Then, there is an $(\epsilon, \delta, \tau)$-tester-learner for $\mathcal{C}$ w.r.t. $\mathcal{D}^*$ with runtime $\text{poly}(A^\ell, (\log(1/\delta))^\ell, (d+1)^\ell, 1/\epsilon)$, and sample complexity at most $\frac{1}{\epsilon^5} \cdot O(Ad)^{2\ell} \cdot \log \frac{1}{\delta}$.*

The following corollary follows from Theorem H.2, combined with the fact that log-concave distributions are hypercontractive (see [Bob01]), as well as the results of [KM13, KKM13] on the sandwiching degree of functions of halfspaces (see Appendix C.2). This is the first result for testably learning even a single halfspace with respect to log-concave distributions up to optimal error.

**Corollary H.3.** *Let $\mathcal{C}$ be the class of arbitrary functions of $k$ halfspaces over $\mathbb{R}^d$ and let $\mathcal{D}^*$ be any log-concave distribution. There is an $(\epsilon, \delta, \tau)$-tester-learner for $\mathcal{C}$ with respect to $\mathcal{D}^*$ that runs in time $(d \log(1/\delta))^{\tilde{O}(\ell)}$, where $\ell = \exp((\log(\log(k)/\epsilon))^{O(k)}/\epsilon^8)$.*

We will now prove Theorem H.2, using once more the iterative polynomial filtering algorithm of Theorem 3.2.

*Proof of Theorem H.2.* The algorithm receives a dataset $\bar{S}_{\text{inp}}$, draws a reference set $S_{\text{ref}}$ of $m_{\text{ref}} = \frac{(C'Ad)^{2\ell}}{\epsilon^5}(\log \frac{1}{\delta})^{4\ell+1}$ i.i.d. unlabeled examples from $\mathcal{D}$, where $C' \geq 1$ is a sufficiently large universal constant and does the following.

1. First, the algorithm runs the filtering procedure of Theorem 3.2 (that is, Algorithm 1) on input $(S_{\text{inp}}, S_{\text{ref}}, m = |S_{\text{inp}}|, \ell, R = \frac{4\tau}{\epsilon} + 2, \epsilon/8)$ to form the filtered dataset $\bar{S}_{\text{filt}}$.

2. Then, the algorithm checks if $m - |\bar{S}_{\text{filt}}| \leq (\tau + \epsilon/2)m$ and rejects if the inequality does not hold.

3. Otherwise, the algorithm finds a polynomial $\widehat{p}$ of degree at most $\ell$ that minimizes the following convex objective.

$$\widehat{p} = \arg\min_p \ \mathbb{E}_{(\mathbf{x},y)\sim \bar{S}_{\text{filt}}}[|y - p(\mathbf{x})|]$$
$$\text{s.t. } p \text{ has degree at most } \ell$$

4. The algorithm outputs $h(\mathbf{x}) = \text{sign}(\widehat{p}(\mathbf{x}) + \widehat{\tau})$, where $\widehat{\tau} \in \mathbb{R}$ minimizes the one-dimensional objective $\mathbb{P}_{(\mathbf{x},y)\sim \bar{S}_{\text{filt}}}[y \neq \text{sign}(\widehat{p}(\mathbf{x}) + \tau)]$ over $\tau \in \mathbb{R}$.

**Soundness.** Suppose, first, that the algorithm accepts. Observe that $|\bar{S}_{\text{inp}} \setminus \bar{S}_{\text{filt}}| = |\bar{S}_{\text{inp}}| - |\bar{S}_{\text{filt}}|$, since $\bar{S}_{\text{filt}} \subseteq \bar{S}_{\text{inp}}$. Since the algorithm has accepted, we know the following.

$$|\bar{S}_{\text{inp}}| - |\bar{S}_{\text{filt}}| = m - |\bar{S}_{\text{filt}}| \leq m(\tau + \epsilon/2)$$

Recall that $\bar{S}_{\text{inp}}$ is a set of $m$ i.i.d. examples from the input distribution $\bar{\mathcal{D}}$. Since $m$ is large enough, due to standard VC dimension arguments (and the fact that $h$ is a polynomial threshold function of degree at most $\ell$), we have that with high probability:

$$\mathbb{P}_{(\mathbf{x},y)\sim\bar{\mathcal{D}}}[y \neq h(\mathbf{x})] \leq \mathbb{P}_{(\mathbf{x},y)\sim\bar{S}_{\text{inp}}}[y \neq h(\mathbf{x})] + \epsilon/4 \tag{H.1}$$

Therefore, it suffices to show a bound on the empirical error of $h$ on $\bar{S}_{\text{inp}}$. We have the following:

$$\mathbb{P}_{(\mathbf{x},y)\sim\bar{S}_{\text{inp}}}[y\neq h(\mathbf{x})] = \frac{1}{m}\sum_{(\mathbf{x},y)\in\bar{S}_{\text{inp}}\setminus\bar{S}_{\text{filt}}}\mathbb{1}\{y\neq h(\mathbf{x})\} + \frac{1}{m}\sum_{(\mathbf{x},y)\in\bar{S}_{\text{filt}}}\mathbb{1}\{y\neq h(\mathbf{x})\}$$

$$\leq \frac{1}{m}|\bar{S}_{\text{inp}}\setminus\bar{S}_{\text{filt}}| + \frac{|\bar{S}_{\text{filt}}|}{m}\mathbb{P}_{(\mathbf{x},y)\sim\bar{S}_{\text{filt}}}[y\neq h(\mathbf{x})]$$

$$\leq \tau + \frac{\epsilon}{2} + \frac{|\bar{S}_{\text{filt}}|}{m}\mathbb{P}_{(\mathbf{x},y)\sim\bar{S}_{\text{filt}}}[y\neq h(\mathbf{x})]$$

It suffices to show that $\mathbb{P}_{(\mathbf{x},y)\sim\bar{S}_{\text{filt}}}[y\neq h(\mathbf{x})] \leq \frac{m}{|\bar{S}_{\text{filt}}|}(\min_{f\in\mathcal{C}}\mathbb{P}_{(\mathbf{x},y)\sim\bar{\mathcal{D}}}[y\neq h(\mathbf{x})] + \epsilon/4)$. Let $f^*$ be the function that minimizes the error on $\bar{\mathcal{D}}$ and let $p_{\text{up}}, p_{\text{down}}$ be its $\frac{\epsilon^2}{128\tau+64\epsilon}$-sandwiching polynomials. Then according to part 2 of Theorem 3.2, and since for $p = p_{\text{up}} - p_{\text{down}}$ we have $\mathbb{E}_{\mathbf{x}\sim\mathcal{D}}[p(\mathbf{x})] \leq \frac{\epsilon^2}{128\tau+64\epsilon}$, we obtain:

$$|S_{\text{filt}}|\mathbb{E}_{\mathbf{x}\sim S_{\text{filt}}}[p_{\text{up}}(\mathbf{x}) - p_{\text{down}}(\mathbf{x})] = \sum_{\mathbf{x}\in S_{\text{filt}}}(p_{\text{up}}(\mathbf{x}) - p_{\text{down}}(\mathbf{x})) \leq \frac{\epsilon m}{8}$$

Therefore, we also have $\mathbb{E}_{\mathbf{x}\sim S_{\text{filt}}}[|f^*(\mathbf{x}) - p_{\text{down}}(\mathbf{x})|] \leq \mathbb{E}_{\mathbf{x}\sim S_{\text{filt}}}[p_{\text{up}}(\mathbf{x}) - p_{\text{down}}(\mathbf{x})] \leq \frac{\epsilon m}{8|S_{\text{filt}}|}$. Observe, now that $h$ is the output of the low-degree polynomial regression algorithm and, hence, according to Theorem A.3 by [KKMS08], we have the following.

$$\mathbb{P}_{(\mathbf{x},y)\sim\bar{S}_{\text{filt}}}[y\neq h(\mathbf{x})] \leq \mathbb{P}_{(\mathbf{x},y)\sim\bar{S}_{\text{filt}}}[y\neq f^*(\mathbf{x})] + \frac{\epsilon m}{8|\bar{S}_{\text{filt}}|}$$

To conclude the argument, observe $\mathbb{P}_{(\mathbf{x},y)\sim\bar{S}_{\text{filt}}}[y\neq f^*(\mathbf{x})] \leq \frac{m}{|\bar{S}_{\text{filt}}|}\mathbb{P}_{(\mathbf{x},y)\sim\bar{S}_{\text{inp}}}[y\neq f^*(\mathbf{x})]$, since $\bar{S}_{\text{filt}} \subseteq \bar{S}_{\text{inp}}$ and, due to a Hoeffding bound, with high probability we have $\mathbb{P}_{(\mathbf{x},y)\sim\bar{S}_{\text{inp}}}[y\neq f^*(\mathbf{x})] \leq \mathbb{P}_{(\mathbf{x},y)\sim\bar{\mathcal{D}}}[y\neq f^*(\mathbf{x})] + \epsilon/8$. Note that due to the choice of $f^*$, we have $\mathbb{P}_{(\mathbf{x},y)\sim\bar{\mathcal{D}}}[y\neq f^*(\mathbf{x})] = \min_{f\in\mathcal{C}}\mathbb{P}_{(\mathbf{x},y)\sim\bar{\mathcal{D}}}[y\neq f(\mathbf{x})] =: \text{opt}$. Overall, we have the following bound:

$$\mathbb{P}_{(\mathbf{x},y)\sim\bar{S}_{\text{inp}}}[y\neq h(\mathbf{x})] \leq \text{opt} + \tau + 3\epsilon/4,$$

which, together with the generalization bound of Eq. (H.1) implies the soundness of the algorithm.

**Completeness.** We will now show that the algorithm accepts with high probability whenever $d_{\text{TV}}(\mathcal{D},\mathcal{D}^*) \leq \tau$. It suffices to show that $|\bar{S}_{\text{inp}}\setminus\bar{S}_{\text{filt}}| \leq (\tau+\epsilon/2)m$ with high probability, since $|\bar{S}_{\text{inp}}\setminus\bar{S}_{\text{filt}}| = m - |\bar{S}_{\text{filt}}|$, which is exactly the quantity based upon which the algorithm decides whether to reject or not.

Let $S_{\text{cln}} = \{\mathbf{x}^{(1)},\mathbf{x}^{(2)},\ldots,\mathbf{x}^{(m)}\}$ be a set of $m$ i.i.d. examples drawn from $\mathcal{D}$. Let $S_{\text{inp}}$ be drawn as follows:

- For each $i\in[m]$, draw an independent Bernoulli random variable $\xi_i$ with probability of success $1-\tau$.

- If $\xi_i = 1$, then let $\mathbf{z}^{(i)} = \mathbf{x}^{(i)}$.

- If $\xi_i = 0$, then draw $\mathbf{z}^{(i)}$ independently from the residual distribution $\mathcal{D}'$ in the maximal coupling between $\mathcal{D}$ and $\mathcal{D}^*$ (i.e., the joint distribution $(\mathbf{x},\mathbf{x}^*)$ that has marginals $\mathcal{D}$ and $\mathcal{D}^*$ respectively and maximizes the likelihood that $\mathbf{x} = \mathbf{x}^*$).

- Let $S_{\text{cln}} = \{\mathbf{z}^{(i)}\}_{i\in[m]}$.

Note that the distribution of each $\mathbf{z}^{(i)}$ is $\mathcal{D}^*$ and they are drawn independently. Due to a Chernoff bound, since $m$ is large enough, we have that, with high probability, $|S_{\text{inp}}\setminus S_{\text{cln}}| \leq m(\tau+\epsilon/8)$. According to Theorem 3.2, we have the following, with high probability:

$$|(S_{\text{inp}}\cap S_{\text{cln}})\setminus S_{\text{filt}}| \leq \frac{1}{R}\cdot|S_{\text{inp}}\setminus S_{\text{filt}}| + \frac{\epsilon m}{8}$$

Observe, now, that $|S_{\text{inp}} \setminus S_{\text{filt}}| = |(S_{\text{inp}} \cap S_{\text{cln}}) \setminus S_{\text{filt}}| + |S_{\text{inp}} \setminus S_{\text{cln}}|$. Therefore, if we solve for $|S_{\text{inp}} \setminus S_{\text{filt}}|$, we have the following

$$\left(1 - \frac{1}{R}\right) \cdot |S_{\text{inp}} \setminus S_{\text{filt}}| \leq |S_{\text{inp}} \setminus S_{\text{cln}}| + \frac{\epsilon m}{8} \leq m(\tau + \epsilon/4)$$

Due to the choice of $R$, we have the desired bound $|\bar{S}_{\text{inp}} \setminus \bar{S}_{\text{filt}}| \leq m(\tau + \epsilon/2)$. $\qquad\square$

