# OpenReview forum: "The Power of Iterative Filtering for Supervised Learning with (Heavy) Contamination"
_NeurIPS.cc/2025/Conference — NeurIPS 2025 spotlight_

### Official Review · Reviewer_bEh6 · 2025-06-27

**Clarity:** 4
**Significance:** 4
**Originality:** 4
**Rating:** 5
**Confidence:** 3

**Summary:**

The paper studies the efficient learnability of families of concept classes and distributions when the training data is adversarially contaminated.
Specifically, in the bounded contamination setting (the "nasty noise" model), a fraction of the training set is removed and replaced by an adversarially chosen set of labeled examples. Notably, the noise affects not only the labels but also the examples (i.e., the covariates).
In this model, the authors present an algorithm that significantly improves the running time for natural concept classes (such as intersections of halfspaces, circuits, etc.) and distributions (such as uniform and Gaussian).
In many cases, the running time matches the best-known results for settings where the contamination affects only the labels.

Moreover, the authors introduce a more challenging contamination model ("heavy contamination"), in which the adversary is allowed to add corrupted points to the training set. As in the previous model, the algorithm achieves non-trivial results, although with increased running time.

Beyond the results themselves, a notable contribution is the algorithmic technique of iteratively removing outliers, applied to concept classes that can be approximated by low-degree polynomials with respect to a hypercontractive distribution.

**Questions:**

For the bounded contamination (BC) setting, you assume that the clean distribution is realizable, whereas in the heavy contamination setting, it can be agnostic. Is there a specific reason why the BC model would not work similarly if the clean distribution were not realizable?

Are the parameters $\eta$ and $Q$ assumed to be known in advance? I couldn't find a clear statement.

Lines 36–37: "all known near-optimal error guarantees for agnostic learning (that can be achieved efficiently) can indeed be extended to the setting of contamination."
In the agnostic model, the results are typically stated with respect to OPT, while in the contamination model the bounds are with respect to 2OPT (i.e., "semi-agnostic"). Is the quoted statement accurate, or did I miss a nuance?

Lines 52–53: "It is known that achieving an error better than twice the contamination rate is, in general, impossible."
Is this impossibility result referring to any algorithm (including inefficient ones)? It might be helpful to clarify.

Table 1: What does the notation $O_k(1)$ mean? Do you mean that the bound hides constants depending on $k$?

**Ethical Concerns:**

["NO or VERY MINOR ethics concerns only"]

**Limitations:**

Limitations are discussed in the paper.

**Quality:**

4

**Strengths And Weaknesses:**

The paper makes strong contributions to efficient supervised learning under contamination, both by improving many known results and by introducing an interesting algorithmic idea.
I believe the paper is a good fit for the conference, and I would be happy to see it accepted.

I did not identify any specific weaknesses.
I am not fully familiar with recent techniques in learning under distribution shifts and unsupervised contamination. If there are similarities with those approaches, it might be helpful for the authors to highlight and discuss them.

---

> ### Author Rebuttal · Authors · 2025-07-30
>
> Thank you for your insightful comments, and for appreciating our work! We will incorporate your feedback in future revisions.
>
> For a comparison to previous outlier removal techniques, see lines 167–178 in the submission, as well as our response to Reviewer b2ov.
>
> **Answers to questions:**
> - The reason we consider the clean distribution to be realizable in the BC setting is to be in line with the classical definition of learning with nasty noise. However, our approach generalizes to agnostic BC learning. See also Definition E.1 in Appendix E, for a formal definition.
> - For BC learning, we do not need to know $\eta$. For HC learning, we do require an upper bound on the contamination ratio $Q$, in order to tune the hyperparameters of the iterative polynomial filtering algorithm. See also Definitions 1.2 and 1.4, where we describe the input of a BC learner (which does not include $\eta$), and the input of an HC learner (which does include $Q$).
> - In lines 36–37, we use the term “optimal” to refer to the information-theoretically optimal guarantee, which, in the BC case, is $2\eta$. Thank you for pointing out this ambiguity. Note that no algorithm (even an inefficient one) can achieve error better than $2\eta$ under bounded contamination in the general case.
> - The notation $O_k(1)$ hides an arbitrary function of $k$, which is a constant when $k$ is considered constant. See also line 775.

---

### Official Review · Reviewer_b2ov · 2025-07-02

**Clarity:** 4
**Significance:** 3
**Originality:** 4
**Rating:** 5
**Confidence:** 2

**Summary:**

This paper introduces a outlier removal algorithm called Interactive Polynomial Filtering and showcases its application in supervised learning with contamination.

The paper also provides guarantees for learning whenever the distributional assumptions are met (hypercontractivity) and: (a) under bounded contamination (BC), guarantee error at most $2\eta + \epsilon$. [Theorem 4.2] (b). under heavy contamination (HC), guarantee error at most  $Q \cdot opt_{total}+\epsilon$. [Theorem 4.4]

**Questions:**

Would the authors mind sharing a quick comparison of their algorithm with the iterative filtering method from [DKS18a]? In particular, what are the main similarities and differences?

**Ethical Concerns:**

["NO or VERY MINOR ethics concerns only"]

**Final Justification:**

I thank the authors for thoroughly addressing my questions. Overall, I think this is a well-written paper and hoping to see it at the conference.

**Limitations:**

yes

**Quality:**

4

**Strengths And Weaknesses:**

Strengths:

1. The paper has some very interesting results and contributions to the field of tolerant testable learning and robust learning with contamination.
2. The paper is well-written and the authors did a great job discussing previous works as well as how this paper fits with related literature.
3. Overall the presentation of the paper is good and the paper was very enjoyable to read :)

Weakness:

I think there is no obvious weakness in general.

---

> ### Author Rebuttal · Authors · 2025-07-30
>
> We wish to thank the reviewer for their positive feedback and comments!
>
> The main differences between our outlier removal algorithm and the one of [DKS18a] are twofold:
>
> 1. First, their algorithm only preserves the expectations of squared polynomials, whereas ours preserves the expectations of arbitrary polynomials with small absolute expectations under the target distribution. Note that a bound on the absolute expectation of a polynomial does imply a bound on its squared expectation via hypercontractivity (see Definition 3.1); however, the resulting bound incurs a multiplicative factor that is exponential in the degree of the polynomial.
>
> 2. Second, our bounds on the polynomial expectations over the filtered set are independent of the degree of the polynomials, while the bounds of [DKS18a] scale exponentially with the degree $\ell$ of the polynomials.
>
> Both of these improvements are crucial in our setting, because the polynomials we are interested in have degrees that are tied to their approximation properties. In particular, applying the results of [DKS18a] directly would introduce a race condition between the approximation quality $\epsilon$ and the degree $\ell$.
>
> See also lines 167–178 in the submission.

---

> > ### Comment · Reviewer_b2ov · 2025-08-04
> >
> > I thank the authors for thoroughly addressing the questions and I will retain my score.

---

### Official Review · Reviewer_Pive · 2025-07-03

**Clarity:** 4
**Significance:** 4
**Originality:** 4
**Rating:** 6
**Confidence:** 3

**Summary:**

The paper presents an outlier removal algorithm called iterative polynomial filtering and its applications to learning under contamination. Specifically, the authors address the nasty noise setting and the heavy contamination setting.

**Questions:**

1. In Table 1, it says that the table has upper and lower bounds, but it is not clear which numbers are which. I suppose that all are upper bounds except for the 2-Heavy contamination result for monotone functions?
2. In practice, how does one select a class of polynomials P? I understand that this is a theoretical paper. However, the algorithm is interesting and it would be nice if there were some discussion on practical implementation.

**Ethical Concerns:**

["NO or VERY MINOR ethics concerns only"]

**Final Justification:**

I thank the authors for their rebuttal, which has addressed the questions raised in my initial review. After consideration, I will keep my score as is.

**Limitations:**

Yes

**Quality:**

4

**Strengths And Weaknesses:**

**Strengths:**

1. The paper is well-written and the mathematics appears to be rigorous. The authors have discussed a good amount of related work.
2. The problem that the paper tries to solve (learning with nasty noise and learning with heavy contamination) is a problem with significant interest in the field. There are many past works on this problem and it is hard, so solving it is meaningful.
3. The approach is applicable to various settings, leading to matching bounds to prior results or extending to novel settings.

**Weaknesses:**

1. The introduction and related work sections are fairly long, going up to page 6. The paper uses terms such as "polynomial p" and "polynomial approximator" early on, but the definitions for these concepts appear quite late in the paper (page 8). It would be helpful for readers if the paper could briefly describe these definitions in the introduction, or at least point to where the definitions can be found.

2. As discussed in the future work section, the filtering algorithm requires access to the clean unlabeled data distribution D*, and the number of samples we need could be large. I am not sure if this is feasible in practical settings. If we have labeled data with some contamination, usually the unlabeled dataset could be contaminated as well. To clean the dataset, one has to know whether a sample is drawn from the target D* or not. If there is an algorithm to do that, then we can also use it to clean the labeled dataset. Otherwise, we may need humans to clean up the dataset, which may not be possible, especially in high-dimensional settings, or it could be expensive. However, as a first step toward solving this problem, this is a minor point.

---

> ### Author Rebuttal · Authors · 2025-07-30
>
> Thank you for your time and appreciation of our work! We are happy to incorporate your suggestions in future revisions of our paper.
>
> **Access to unlabeled examples:** We would like to point out that one does not always need to know whether the clean distribution corresponds to the target $D^* $, at least in the bounded contamination case. In particular, we may first fix a target distribution of interest $D^* $ of our own choosing, and form an algorithm that is guaranteed to either produce a low-error hypothesis under the clean distribution, or certify that the clean distribution is far from $D^* $. In other words, this algorithm would work in the testable setting, meaning that it does not require any assumptions on the clean distribution, at the cost of potentially refusing to provide a solution, but only if the clean distribution happens to be far from $D^* $.
>
> To achieve this type of guarantee, it suffices to raise a flag if the outlier removal with respect to $D^* $ removes more than an $O(\eta)$ fraction of the input examples. Since $D^* $ is chosen by the learner, we can generate our own samples, for example by using random coin flips.
>
> The proof of correctness for this method follows the approach of Section H, where we show that our iterative polynomial filtering can be used to solve the tolerant testable agnostic learning problem. We did not define the testable version of BC-learning for clarity of presentation, but the proofs of Section H can be generalized to the BC setting as well.
>
> Nevertheless, we agree that designing algorithms that work universally with respect to wide families of distributions (and, hence, lead to testable algorithms that accept more often) is a concrete and important direction for future work.
>
>
> **Answers to questions:**
>
> **Q1.** You are correct that the only lower bound in Table 1 is in the cell of the last row and last column. We will make this clear in future revisions.
>
> **Q2.** The choice of the class of polynomials is characterized by two hyperparameters, the degree $\ell$ and the excess error rate $\epsilon$. If one is interested in achieving information-theoretically optimal guarantees (e.g., error $2\eta$ in the BC case), then the choice of $\ell$ should be as high as possible, depending on the computational budget of the learner. The hope would be that $\ell$ is large enough so that the ground truth can be approximated by degree-$\ell/2$ polynomials.
>
> Moreover, the error of the output hypothesis on the filtered set is an accurate proxy for its error on the clean distribution, as long as the number of removed points during filtering is small. Both of these quantities can be estimated efficiently, and, therefore, one can verify whether the choice of $\ell$ is large enough.
>
> On the other hand, if the learner can only afford small values for $\ell$ (e.g., $\ell=1$ or $2$), then it would be preferable to follow approaches of prior work on BC learning (see, e.g., [DKS18a], citation in lines 481–483). These works give guarantees for BC learning of geometric classes (like halfspaces or intersections thereof), through constant-degree polynomial filtering. For a comparison between our filtering algorithm and the one used in [DKS18a], see our response to Reviewer b2ov.
>
> The caveats are that (1) these methods can be significantly less robust to contamination compared to ours and (2) they only seem to work for special classes. Therefore, their methods would only provide meaningful results if the contamination rate is small enough, and the ground truth lies within a structured concept class. One interesting direction for future work is extending these methods to the case of heavy contamination, when $\mathrm{opt}_{\mathrm{total}}$ takes small values.

---

### Official Review · Reviewer_E9uM · 2025-07-07

**Clarity:** 3
**Significance:** 3
**Originality:** 3
**Rating:** 5
**Confidence:** 2

**Summary:**

They propose efficient supervised learning algorithms under contamination where both labels and covariates are adversarialy corrupted. Here, a learner has access to a potentially heavy contaminated dataset and the goal is to learn a model that performs well on some clean underlying distribution.

They propose an outlier removal algorithm called iterative polynomial filtering in the supervised setting.
	-  They show any function that can be approximated by low-degree polynomials can be efficiently learned under bounded noise contamination. Here, the learner receives a dataset with bounded contamination and the goal is to output a classifier that enjoys approximately optimal error guarantees on the clean underlying target distribution.

	This solves a long-standing gap between the complexity of agnostic learning and learning w contamination.

	- If a function class admits stronger notion of sandwiching approximators, they obtain near-optimal learning guarantees even in the heavy-contamination case in the case of supervised learning binary classifiers.



For learning with bounded contamination (BC-learning), their algorithm runs in two phases: 1- run an outlier removal algorithm. 2. construct a predictor based on the polynomial \hat{p} with smallest \ell_1 error on the filtered set from step 1.

The main idea of their alg is the outlier removal algorithm and it comes from bounded-degree outlier removal procedures from robust learning and learning with distribution shift. The idea is that one can iteratively find polynomials that violate the desired condition over the input set and use them to filter the input points. More specifically, the algorithm removes the points that give such polynomials values larger than a threshold. After this step, they can ensure the outliers remaining are not dangerous when moving to step 2.

**Questions:**

Table 1, is the caption as intended? I am confused which one is the lower bound and which one is the upper bound?

In Alg 1, what is the size of \cal{P}? In line 5, how can you compute p* efficiently?

Their algorithms require sample access to the target distribution D^*. What is the main challenge in order to relax this assumption?

**Ethical Concerns:**

["NO or VERY MINOR ethics concerns only"]

**Final Justification:**

After going through the rebuttal and other responses, I agree that this is a strong set of results and would be happy to see it gets accepted.

**Limitations:**

yes

**Paper Formatting Concerns:**

no concerns

**Quality:**

3

**Strengths And Weaknesses:**

Strengths: I am mostly familiar with robust learning (when the features do not change) and outlier detection in unsupervised learning and not very familiar with this line of work. Their problem is challenging since both labels and covariates can be adversarially corrupted. Furthermore, the heavy-contamination results are also interesting and they also build-up on their filtering algorithm. They propose a new model for learning binary classifiers with heavy additive contamination that outputs a single hypothesis with a strong error guarantee under the clean distribution.

In terms of weaknesses, the runtime of their algorithm is pretty huge, but the gap between lower and upper bound is not too large (table 3).

---

> ### Author Rebuttal · Authors · 2025-07-30
>
> We thank the reviewer for their constructive feedback and for appreciating our work.
>
> In table 1, we present the upper bounds using big-$O$ and the lower bound for monotone functions (last row, last column) using $\Omega$. We will modify the caption to make this clear in future revisions.
>
> Computing $p^*$ corresponds to a convex program over the space of coefficients of degree-$\ell$ polynomials. In particular, the objective function is linear, and $\cal{P}$ is a convex set defined by a polynomial number of linear constraints plus one quadratic constraint. See lines 869–873 in the submission (proof of Theorem 3.2) for a relevant discussion on the algorithmic implementation.
>
> We believe that obtaining an outlier-removal algorithm that works universally over a wide family of distributions is an important direction for future work (see lines 356–360). There are several challenges towards achieving this:
> 1. First, the problem is non-trivial even from an information-theoretic perspective. Barring any computational considerations, one needs to be able to control the number of clean points that are removed by the filtering algorithm. However, it is unclear when this is possible for infinite-sized families of target distributions (e.g., the family of all isotropic log-concave distributions).
> One special case where this issue can be resolved is for concept classes with the following universal approximation property: for each function in the class, there is a low-degree approximator that works with respect to all distributions within a wide family.
>
>
> 2. Second, even if one could show such a universal approximation property, it is not clear how to express the universal approximation (or an appropriate relaxation thereof) as a convex constraint. To this end, the sum-of-squares framework could be useful, but none of the known results from the sum-of-squares literature seems to directly apply to our setting.

---

> ### Comment · Reviewer_E9uM · 2025-08-04
>
> Thank you for your response. I remain convinced that this is a strong set of results and retain my score.

---

### Decision · Program_Chairs · 2025-09-17

**Decision:**

Accept (spotlight)

**Comment:**

The authors consider the problem of efficient supervised learning with nasty noise. This problem is less studied in the literature (unlike supervised learning under label noise or unsupervised learning with adversarial contamination) but is quite important and interesting. Under distributional assumptions (such as Gaussian marginal, etc) and for certain interesting concept classes related to polynomials, the authors propose algorithms for learning under bounded contamination. These results match those of the easier model of label noise. The authors also go beyond this, and leverage the testable learning framework to make sure that the distributional assumptions are actually met.

This work is inspired by recent line of work on testable learning under distribution shift where the authors propose an iterative filtering approach to learning with contamination. They use this algorithm to show multiple results including (i) the first efficient algorithms for tolerant testable learning of halfspaces with respect to a log-concave distribution and (ii) an efficient method to learn any function class that can be approximated by low-degree polynomials with respect to a hypercontractive distribution under nasty noise.

This work offers a fresh view on the important problem of robust supervised learning, offering algorithmic and theoretical contributions on this important problem. The work would likely have significant impact on the theoretical ML community.